# Bacterial Adaptation to Venom in Snakes and Arachnida

Elham Esmaeilishirazifard,[a,b]* Louise Usher,[a,b] Carol Trim,[c] Hubert Denise,[d§] Vartul Sangal,[e] Gregory H. Tyson,[f] Axel Barlow,[g] Keith F. Redway,[a] John D. Taylor,[a,b,h] Myrto Kremyda-Vlachou,[a] Sam Davies,[e] Teresa D. Loftus,[i] Mikaella M. G. Lock,[i] Kstir Wright,[a] Andrew Dalby,[a] Lori A. S. Snyder,[j] Wolfgang Wuster,[k] Steve Trim,[i] Sterghios A. Moschos[a,b,e]

[a]Department of Biomedical Sciences, Faculty of Science and Technology, University of Westminster, London, United Kingdom
[b]Westminster Genomic Services, Faculty of Science and Technology, University of Westminster, London, United Kingdom
[c]School of Psychology and Life Sciences, Faculty of Science, Engineering and Social Sciences, Canterbury Christ Church University, Canterbury, Kent, United Kingdom
[d]EMBL-EBI European Bioinformatics Institute, Wellcome Trust Genome Campus, Hinxton, Cambridge, United Kingdom
[e]Department of Applied Sciences, Faculty of Health and Life Sciences, Northumbria University, Newcastle, Tyne and Wear, United Kingdom
[f]Food and Drug Administration, Center for Veterinary Medicine, Office of Research, Laurel, Maryland, USA
[g]Institute for Biochemistry and Biology, University of Potsdam, Potsdam, Germany
[h]School of Environment and Life Sciences, University of Salford, Salford, Greater Manchester, United Kingdom
[i]Venomtech, Ltd., Sandwich, Kent, United Kingdom
[j]School of Life Sciences, Pharmacy, and Chemistry, Kingston University, London, United Kingdom
[k]Molecular Ecology and Evolution at Bangor, School of Biological Sciences, College of Natural Sciences, Bangor University, Bangor, Wales, United Kingdom

Elham Esmaeilishirazifard and Louise Usher contributed equally to this article. Author order was determined on the basis of seniority.

**ABSTRACT** Animal venoms are considered sterile sources of antimicrobial compounds with strong membrane-disrupting activity against multidrug-resistant bacteria. However, venomous bite wound infections are common in developing nations. Investigating the envenomation organ and venom microbiota of five snake and two spider species, we observed venom community structures that depend on the host venomous animal species and evidenced recovery of viable microorganisms from black-necked spitting cobra (*Naja nigricollis*) and Indian ornamental tarantula (*Poecilotheria regalis*) venoms. Among the bacterial isolates recovered from *N. nigricollis*, we identified two venom-resistant, novel sequence types of *Enterococcus faecalis* whose genomes feature 16 virulence genes, indicating infectious potential, and 45 additional genes, nearly half of which improve bacterial membrane integrity. Our findings challenge the dogma of venom sterility and indicate an increased primary infection risk in the clinical management of venomous animal bite wounds.

**IMPORTANCE** Notwithstanding their 3 to 5% mortality, the 2.7 million envenomation-related injuries occurring annually—predominantly across Africa, Asia, and Latin America—are also major causes of morbidity. Venom toxin-damaged tissue will develop infections in some 75% of envenomation victims, with *E. faecalis* being a common culprit of disease; however, such infections are generally considered to be independent of envenomation. Here, we provide evidence on venom microbiota across snakes and arachnida and report on the convergent evolution mechanisms that can facilitate adaptation to black-necked cobra venom in two independent *E. faecalis* strains, easily misidentified by biochemical diagnostics. Therefore, since inoculation with viable and virulence gene-harboring bacteria can occur during envenomation, acute infection risk management following envenomation is warranted, particularly for immunocompromised and malnourished victims in resource-limited settings. These results shed light on how bacteria evolve for survival in one of the most extreme environments on Earth and how venomous bites must be also treated for infections.

**KEYWORDS** drug resistance evolution, extremophiles, genome analysis, microbiome, multidrug resistance, venom

Address correspondence to Sterghios A. Moschos, sterghios.moschos@northumbria.ac.uk.
*Present address: Elham Esmaeilishirazifard, Cancer Research UK Cambridge Institute, University of Cambridge, Cambridge, United Kingdom.
§Present address: Hubert Denise, UK Health Security Agency, London, United Kingdom.

The authors declare no conflict of interest.

The rise of multidrug-resistant (MDR) bacterial infections suggests the end of the antibiotic golden era might be approaching fast. Discovery of novel antimicrobials is therefore an urgent priority of exceptional socioeconomic value. Crude preparations of animal venoms exhibit strong antibiotic potencies, including against clinical MDR bacterial isolates such as *Mycobacterium tuberculosis* (1). With antimicrobial properties described for crotalid (pit viper) venom as early as 1948 (2), relevant compounds have been isolated from most animal venoms, including those of spiders, scorpions, and insects, as well as aquatic species. Examples include phospholipase A2 (PLA2) enzymes, L-amino acid oxidases, cathelicidins, C-type lectins, and hydrophobic/cationic peptides, as well as venom toxin domains (3), which may act by physically disrupting bacterial cell membranes through pore formation (4, 5). Accordingly, venomous animal bite or sting (envenomation) wound infections are considered rare (6) and are attributed to secondary infection (7). Yet, over three-quarters of snake bite victims may develop mono- or polymicrobial envenomation wound infections, characterized by *Bacteroides*, *Morganella*, *Proteus*, and *Enterococcus* (8, 9)—bacterial taxa commonly found in the gut. Indeed, *Enterococcus faecalis* and *Morganella morganii* have been independently reported as the most common Gram-positive and Gram-negative envenomation wound infections across several countries (8–10). Historically associated with the oral snake microbiota (11), these bacteria are thought to originate from prey feces (12) persisting in the snake oral cavity (10), with a diversity similar to that of the snake gut (13). Yet, no "fixed" oral microbiota were observed in early systematic studies, beyond a seasonal variation of diversity (14). Curiously, mouths of nonvenomous snakes were reportedly more sterile than those of venomous snakes (14), a counterintuitive finding independently reproduced elsewhere (10). More recently, the oral microbiota of the nonvenomous Burmese python (*Python bivittatus*) has also been reported to be native and not derived from prey guts (15).

As venom glands are connected to the tip of envenomation apparatus via a persistently open duct that is continuously exposed to the environment (16), the envenomation apparatus could be compared to clinical catheterization assemblies: a transcutaneous needle resting on a nonsterile environment, connected to a continually open duct, leading to a liquid-containing vessel. Such devices develop biofilms within a few days, making weekly catheter replacement necessary (17). Unlike the high flow rates of catheters, however, the envenomation apparatus normally ejects venom only sporadically. Captive snakes are often fed weekly and can fast for months, whereas large arachnids typically are fed monthly. Wild animals may also undergo hibernation for several months, when venom expulsion frequency can be assumed to be zero. Collectively, these conditions offer opportunities for microbes to colonize venom across its concentration gradient from the animal mouth to the venom gland, potentially driving the evolution of resistance against venom antimicrobials.

The aims of this study were to evaluate whether the microbiota of venom differs from that of the envenomating apparatus, whether microbes can survive in venom, and, if so, what genetic adaptations facilitate their survival in such extreme microenvironments. Using culture and culture-free methods, we examined this hypothesis in snakes and spiders and then employed antimicrobial susceptibility testing, whole-genome sequencing analysis, and comparative genomics to identify and characterize venom tolerance and infectivity potential in representative novel *E. faecalis* isolates from venom from captive *Naja nigricollis* (black-necked spitting cobra).

## RESULTS

**The snake venom microbiota vary on account of host species and not on account of the oral microbiota.** Applying established culture-free methods (18) on commercially available venom from *Bothrops atrox* (fer-de-lance; Viperidae) and a venom sample from a captive *Bitis arietans* (African puff adder), we first optimized microbial DNA extraction for this unusual biological matrix (see Fig. S1 in the supplemental material). Given animal availability, behavioral, and sampling limitations, such as the limited and inconsistent venom yields of scorpions, we next focused on five snake and two spider

**TABLE 1** Animals sampled for the presence of microbiomes in venom

| Common name | Scientific name | Short name | Origin | Preservation method | No. of animals |
|---|---|---|---|---|---|
| Snakes | | | | | |
| Puff adder | *Bitis arietans* | *B. are* | Captivity | Flash frozen | 1 |
| | | L | Commercial | Lyophilized | 1 |
| | | B1–B8 | Wild | Air dried | 8 |
| Black-necked cobra | *Naja nigricollis* | *N. nig* | Captivity | Flash frozen | 3 |
| Fer-de-lance | *Bothrops atrox*[a] | *B. atr* | Captivity | Flash frozen | 3 |
| Western diamond rattlesnake | *Crotalus atrox*[a] | *C. atr* | Captivity | Flash frozen | 2 |
| Taipan | *Oxyuranus scutellatus*[a] | *O. scu* | Captivity | Flash frozen | 2 |
| | | | | | |
| Spiders | | | | | |
| Indian ornamental | *Poecilotheria regalis* | *P. reg* | Captivity | Flash frozen | 3 |
| Salmon pink | *Lasiodora parahybana*[b] | *L. par* | Captivity | Flash frozen | 5 |

[a]Venom produced by one animal only.
[b]Yields ranged from <1 to 30 $\mu$L.

species (Table 1). We collected a swab (sample O) of the oral cavity (snakes) or fang surface (spiders) and two consecutive envenomation samples (E1 and E2), expecting the second venom sample to have fewer contaminants from bacterial plugs possibly forming on the envenomation apparatus. In agreement with previous reports (11, 14), principal-coordinate analysis (PCoA) and unsupervised clustering (Fig. S2) failed to discriminate the swab microbiota by host species, suggesting the common diets and, most likely, water sources in captivity had the biggest impact.

This led us to hypothesize that captive animals would feature more closely related microbiota in their venoms than commercial or wild samples. We therefore compared the venom microbiota of all snakes using the same approach (Fig. 1). High Shannon-Wiener indices indicated considerable diversity in snake venom microbiota; however, surprisingly closer relationships were observed between *B. arietans* and other Viperidae, despite samples spanning captive and wild animals; an exception was *B. atrox* venom, which was characterized principally by *Gammaproteobacteria*. Focusing on *B. arietans* also failed to cluster samples by origin (Fig. S3), despite the unknown providence of the commercially available lyophilized sample and the disparate locations across South Africa where wild *B. arietans* samples were collected (Fig. 1D); among the latter, the air-drying method used for venom preservation could have been expected to substantially compromise the microbiota signature. In contrast, *N. nigricollis* microbiota largely formed a distinct cluster (Fig. 1) characterized by *Bacteroidia* (*Bacteroidaceae*), a taxon less common among Viperidae. This could reflect anatomical differences in elapid (cobra) fang location at the front of the mouth compared to the sheathed nature of the longer, hinged viperid fangs, whose tips rest at the back of the oral cavity. In contrast, spider species did not seem to influence venom microbiota consistency and exhibited lower biodiversity (Fig. S4). These results likely reflected vertebrate/invertebrate anatomical differences and the limited venom yield from invertebrates (<1 to 30 $\mu$L) versus snakes (100 to 1,000 $\mu$L).

**A fifth of the *N. nigricollis* venom microbiota is distinct from that of fangs.** Encouraged by the distinctive bacterial taxonomies in *N. nigricollis* venom, the availability of animals under controlled conditions, and the paired nature of the fang swab and envenomation samples, we delved deeper into this data set. The fang microbiota appeared to form a cluster distinct from that of venom microbiota (Fig. 2A and B), suggesting the venom gland might be a distinct ecological niche (Fig. 2C). We therefore asked if any bacterial taxa were unique to subsets of these samples. Operational taxonomic unit (OTU) incidence analysis within each animal (Fig. 2D) suggested some 60% of OTUs were shared between corresponding venoms and fangs. Yet, importantly, up to 20% of these appeared to be unique to venom, and some 15% were unique to the fang (Fig. 2E), indicating an OTU continuum between the two microenvironments, with unique taxa in each site. Common sample types also featured a majority of common taxa and OTUs unique to each site in each animal (Fig. 2F). However, taxa unique to

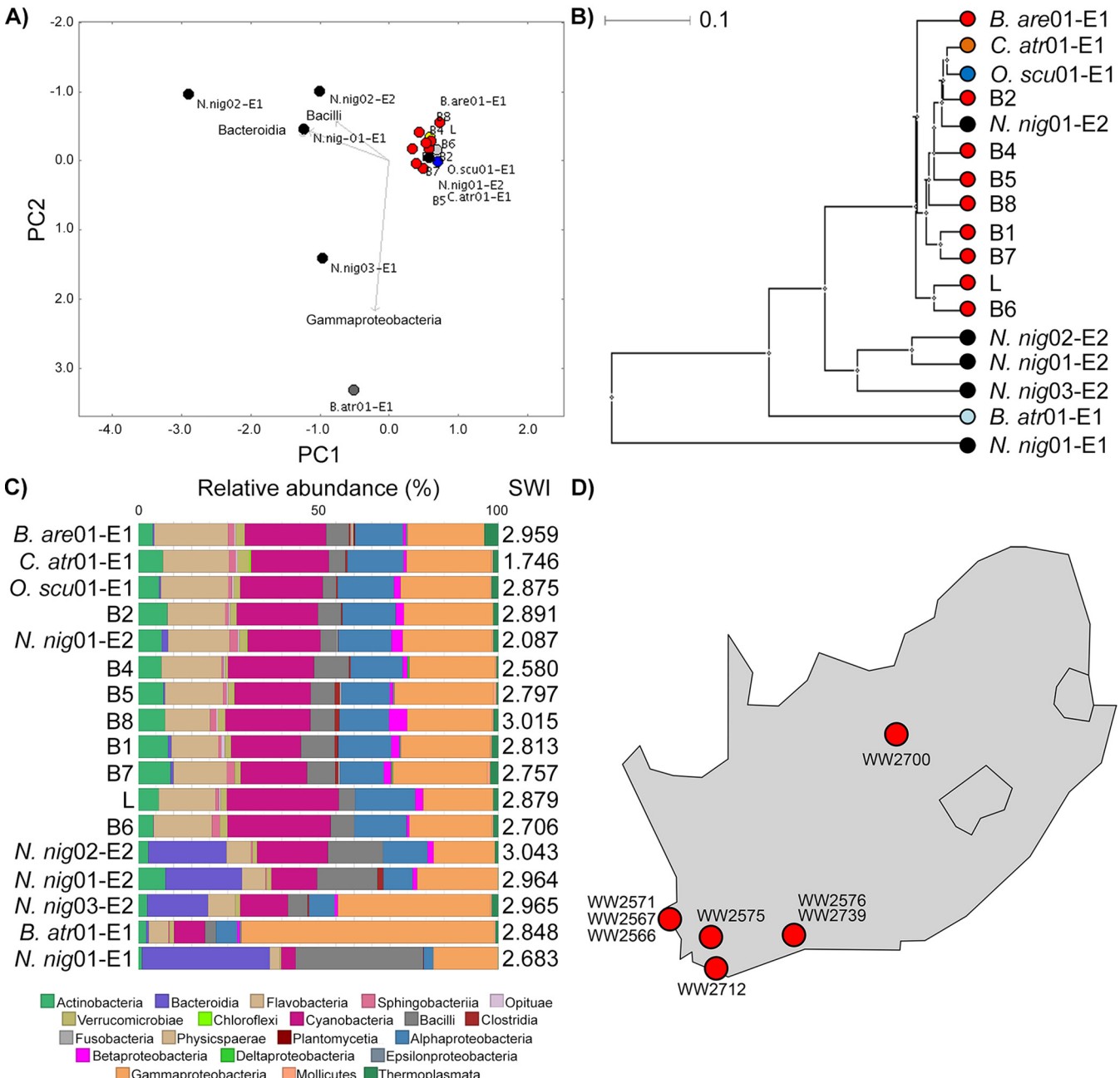

**FIG 1** Snake venom microbiomes cluster on account of host species. Viperid venom microbiomes cluster separately from *N. nigricollis*, with the exception of *B. atrox*, as determined by (A) PCoA, (B) an UPGMA tree (i.e., unweighted pair group method using average linkages), and (C) class-level taxonomic profiling following 16S rRNA phylogenetic analysis. Dots in panels A and B are colored by species (red, *B. arietans*; black, *N. nigricollis*; light blue, *B. atrox*; orange, *C. atrox*; dark blue, *O. scutallatus*), represent data from individuals in captivity, are labeled with the short species name, are enumerated for the individual, and are identified for the envenomation number (E1 or E2) of the sample. The 8 wild *B. arietans* samples (red dots B1 to B8) and the commercially sourced, lyophilized *B. arietans* sample (red L dot) are independently labeled. Sample B3 was removed from the analysis due to the yield of ∼100× lower read depth from this sample compared to all other *B. arietans* samples. Relative taxonomic diversity profiles in panel C are aligned to the UPGMA tree sample labels, with the Shannon-Wiener index (SWI) of each sample indicated. The geographical origins of the wild *B. arietans* samples collected in South Africa are shown in panel D.

each sample type (O, E1, or E2) were rarely found across all animals. These results suggested that although the microbiota between each snake fang and venom were largely common, venom contained distinct organisms.

**The venom microbiota in snakes and spiders is viable.** After identifying all cultivable and noncultivable bacterial species in swabs and venoms, we next proceeded to examine if cultivable aerobes could be recovered from these samples, as an indication of adaptation to venom. Testing for microbial viability (Table S1) yielded less growth

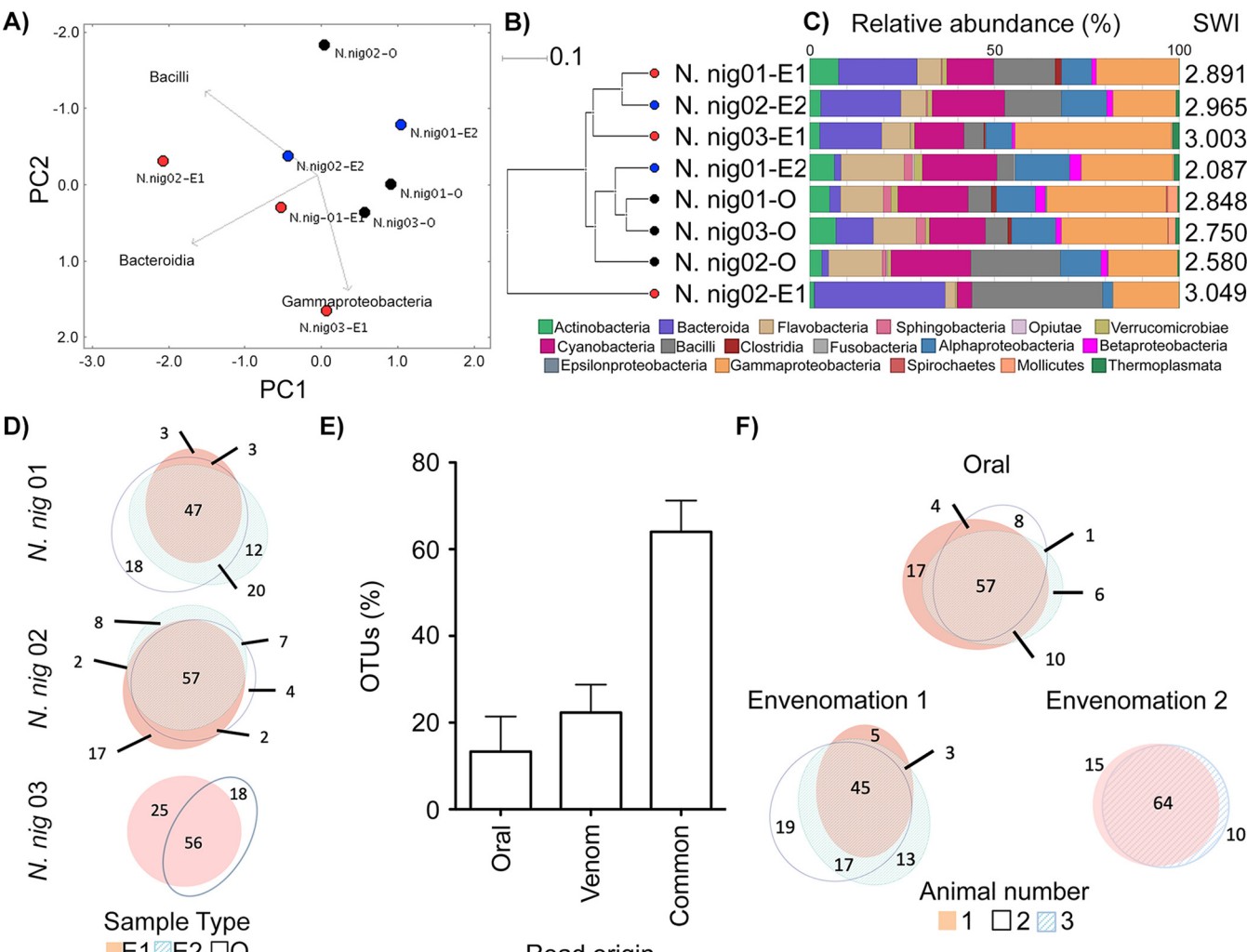

**FIG 2** The intra- and interindividual relationships of venom and oral microbiomes in *N. nigricollis*. Comparison of the oral and venom microbiomes in three *N. nigricollis* individuals by (A) PCoA, (B) UPGMA tree, and (C) class-level taxonomic profiling following 16S rRNA phylogenetic analysis indicates separate clustering of the microbiota in the two microenvironments. (D) Within-animal incidence comparisons of OTUs suggest (E) unique taxa exist within the oral but also the venom microenvironments. (F) Between-animal comparisons per niche (E1, E2, and oral) indicate most OTUs are shared, but some OTUs are unique to each animal for each site. Dots in panels A and B represent individual *N. nigricollis* (*N. nig*) animal data and are colored/labeled by sample type (black, oral; red, envenomation 1 [E1]; blue, envenomation 2 [E2]). Relative taxonomic diversity profiles in panel C are aligned to the UPGMA tree sample labels, with the Shannon-Wiener index (SWI) of each sample indicated. The "venom" histogram in panel E represents the sum OTU fraction found in the two envenomation samples per individual (± standard deviation).

with swab samples. Where this was significant, it was not usually matched by similar growth from the corresponding venom samples, further suggesting that the venom bacteria were probably not mouth contaminants. Strikingly, substantial and consistent growth was encountered among *N. nigricollis* (Fig. 3A) and *P. regalis* (Table S1) samples on blood agar. Unexpectedly, neither the wild (air-dried) nor the commercial (lyophilized) venom samples yielded any growth, although colonies were obtained in blood agar from the captive *B. arietans* fangs, underscoring the impact of venom handling on microbial viability, at least for aerobic bacteria.

Clinical biochemistry tests identified the multiple, punctate white colonies from *N. nigricollis* almost universally as *Staphylococcus* spp., albeit with assay confidence intervals (CIs) below 50% (Table S2). In contrast, *Stenotrophomonas maltophilia* (80.4% CI) was present in five out of six *Poecilotheria regalis* (all animals positive) and two *Lasiodora parahybana* (salmon pink tarantula) venom samples, but not on any fang swabs. Perplexed by the *N. nigricollis* results, we sequenced these isolates on the Ion Torrent Personal Genome Machine (PGM).

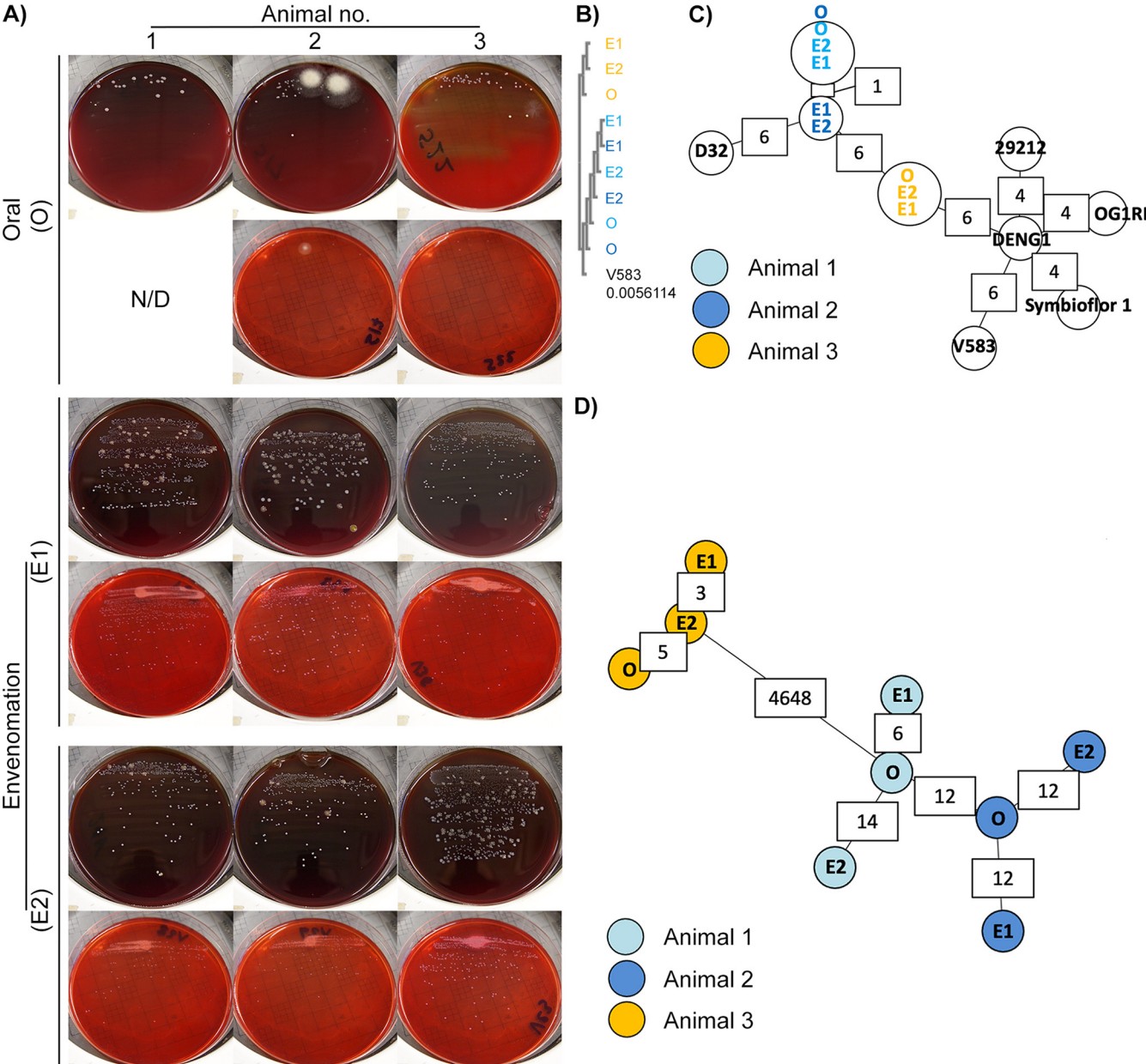

**FIG 3** Whole-genome sequencing identifies viable bacteria in *N. nigricollis* venom as two animal-specific *E. faecalis* strains. (A) White punctate colonies were recovered in blood agar (upper panels) and MacConkey agar (lower panels) blind cultures of individual oral swab (O) and two consecutive envenomation samples (E1 and E2) obtained from three captivity *N. nigricollis* snakes. N/D, none detected. (B) Blind multiple-sequence alignment (ClustalO followed by ClustalW phylogeny) of homologous sequences across the *de novo* assembled genomes against the *E. faecalis* V583 *katA* gene (distance to V583 *katA* indicated in the V583 track) suggests two sequence groups reflecting the history and housing of the host animals. (C) Blind MST construction based on the MLST of the *N. nigricollis*-derived isolates against nine *E. faecalis* reference genomes again separates samples into two distinct clusters that reflect the history and housing of the host animals. Partially available allele data are included in this analysis, and instances of allelic differences between nearest neighbors are annotated in white boxes. (D) Blind complete genome MLST against a custom schema generated using three closely related *E. faecalis* reference genomes clusters these isolates by animal of origin (animal 1, light blue; animal 2, dark blue; animal 3, orange). The host animal color scheme depicted in panel D is also used in panels B and C.

**Viable bacteria in *N. nigricollis* venom are two new *E. faecalis* sequence types.** Resequencing against putative reference genomes (Table S2) demonstrated less than 6% base alignment across all isolates. Instead, BLASTn analysis of the largest *de novo* assembled contigs identified *E. faecalis* V583 as the closest likely relative (>80% base alignment, 51.2× coverage). This microbe is considered of mammalian origin and is usually found in soils, waters, and foodstuffs probably arising from mammalian gastro-intestinal tracts via feces. This was puzzling given the catalase-positive biochemistry of

**TABLE 2** Novel sequence types of *E. faecalis* recovered from fangs and venoms of *N. nigricollis*

| | Sample | | Locus | | | | | | |
| Animal no. | Isolate origin[a] | Blinding code | *gdh* | *gyd* | *pstS* | *gki* | *aroE* | *xpt* | *yqiL* |
|---|---|---|---|---|---|---|---|---|---|
| 1 | O | S22 | 22 | 6 | 31 | 13 | 11 | 35 | 8 |
| | E1 | V36 | 22 | 6 | 31 | 13 | 11 | 35 | 8 |
| | E2 | V29 | 22 | 6 | 31 | 13 | 11 | 35 | 8 |
| | | | | | | | | | |
| 2 | O | S17 | 22 | 6 | 31 | 13 | 11 | 35 | 8 |
| | E1 | V31 | 22 | 6 | 31 | 13 | 11 | 35 | 8[b] |
| | E2 | V28 | 22 | 6 | 31 | 13 | 11 | 35 | 8[c] |
| | | | | | | | | | |
| 3 | O | S3 | 18 | 1 | New | 24 | 83 | 47 | New |
| | E1 | V33 | 18 | 1 | New | 24 | 83 | 47 | New |
| | E2 | V23 | 18 | 1 | New | 24 | 83 | 47 | New |

[a]O, oral swab sample; E1, envenomation 1; E2, envenomation 2.
[b]Single-base-pair deletion in NGS data not validated by Sanger sequencing.
[c]Homopolymer single-base extension not validated by Sanger sequencing.

the isolates versus the generally accepted catalase-negative nature of *E. faecalis* (19), but was explained by confirming the *E. faecalis* V583 *katA* gene, coding for a heme-dependent cytoplasmic catalase (20), among all isolates at 99% identity. Blind multiple-sequence alignment (MSA) further revealed two *katA* alleles: one among isolates from animals 1 and 2 (allele 1) versus another found in animal 3 isolates (allele 2) (Fig. 3B) differing from the V583 allele by less than 20 single nucleotide polymorphisms (Fig. S5A). Interestingly, these alleles grouped isolates according to the origin and joint housing histories of animal 3 (group A) versus animals 1 and 2 (group B).

To explore isolate relationships further, we generated minimum spanning trees (MSTs) (Fig. 3C) by multilocus sequence typing (MLST) (Table 2), including at core genome level (cgMLST) (Fig. 3D; Fig. S5B to D) (21, 22). Comparisons to five complete *Enterococcus faecium* genomes succeeded only for the *gyd* (alelles 16 and 19) and *adk* (allele 18) loci. In contrast, MLST succeeded for all *E. faecalis* loci (Table 2), reinforcing *katA* allele observations (Fig. 3B) and identifying two novel sequence types featuring two new alleles for *pstS* and *yqiL* (Fig. S6), as confirmed by Sanger sequencing. MLST also indicated closer relationships to the *E. faecalis* strains OG1RF (87.5% ± 1.7% of OG1RF cgMLST targets), D32 (78.8% ± 2.1%), and DENG1 (77.4% ± 1.5%). Pairwise comparisons of the resulting custom cgMLST schema, including 5,041 loci found across all the *N. nigricollis*-derived isolates, further reinforced isolate grouping (Fig. 3C), suggesting two independent *E. faecalis* strain acquisition events across these three animals.

**Pangenomic and experimental evidence of *E. faecalis* isolate adaptation to venom.** Expanding cgMLST by an additional 3,060 loci found in some *E. faecalis* isolates (Fig. S5D) identified 290 to 831 allelic differences within each animal. While 80.9% of alleles varied between the two nearest-neighbor isolates from the two strains, venom isolates from group B animals were divergent by 7.15 to 10.3% from their oral isolate counterparts, indicating that genomic changes were occurring with increased frequency, possibly in response to selective pressure applied by the venom. Given the well-described plasticity of the *E. faecalis* genome (OG1RF versus V583, 2.74 versus 3.36 Mb), we next examined mobile element divergence as the potential source of this adaptation. Detection of the pTEF2 gene *repA-2* (plasmid initiator protein) (Table S3) suggested only plasmid fragments were found in these genomes. As pTEF2 is one of three *E. faecalis* V583 plasmids associated with vancomycin resistance (23), we next confirmed the presence of fragments from the other pTEF plasmids in patterns consistent with the isolate groupings (Table S4), and some sequences absent from group A secondary envenomation isolates (Fig. 4A). As many of the genomic elements with high (>95%) sequence identity to these plasmids were known, highly mobile sequences (e.g., the *E. faecalis* Bac41 bacteriocin locus) (24), these results suggested their participation in the genomic divergence of *E. faecalis* within each animal. Moreover,

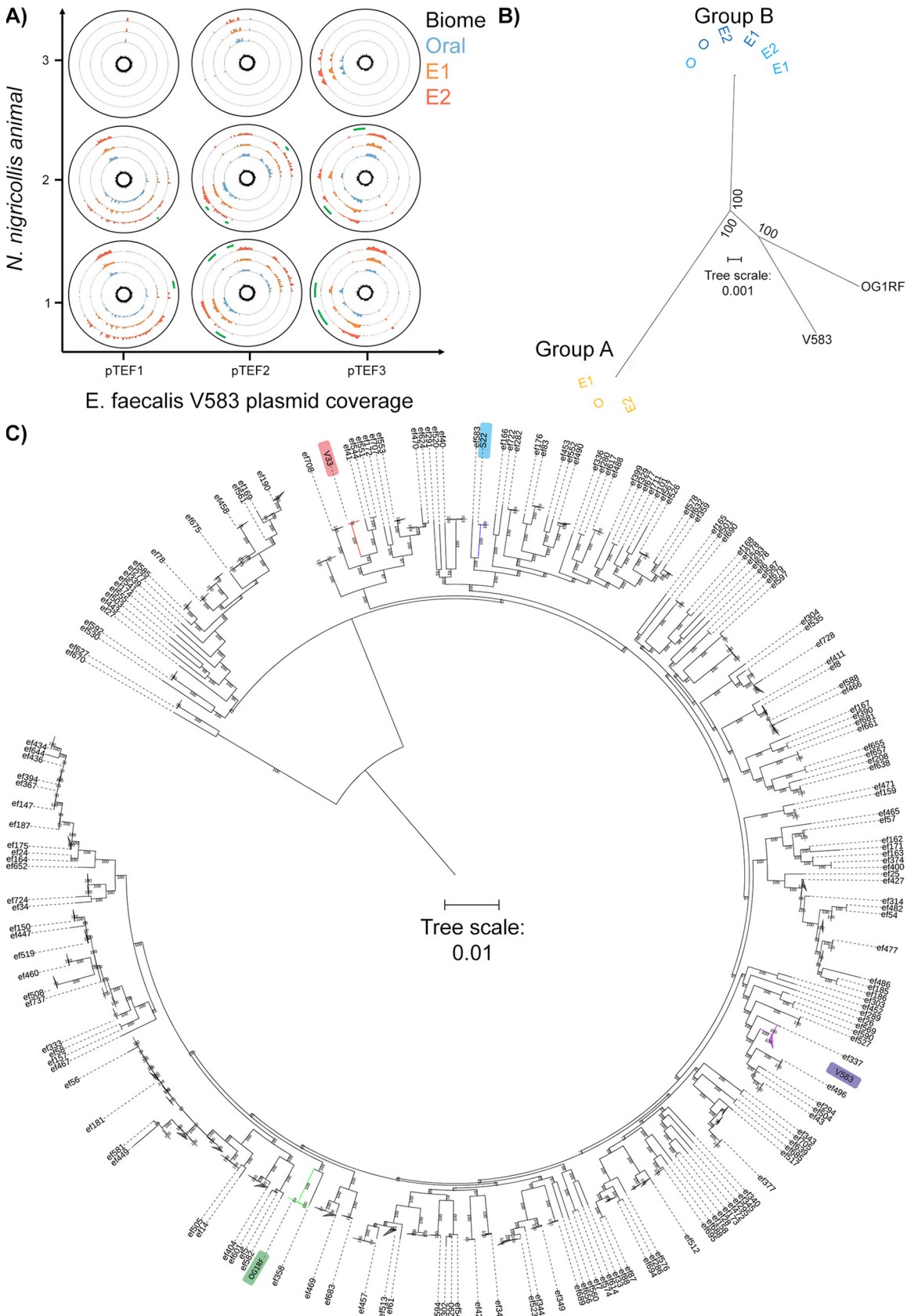

**FIG 4** Comparative genomics of mobile and core genomic chromosomal elements of venom-tolerant *E. faecalis*. (A) Circos coverage plots of the vancomycin resistance-associated V583 plasmids pTEF1, pTEF2, and pTEF3 in the *E. faecalis* isolates obtained from oral,

prompted by the detection of bacteriocin 41, a search for additional bacteriocins revealed a distinct cadre of genes among the two isolate groups. Thus, group A featured a class II bacteriocin, MR10A, MR10B, and enterolysin A, while group B shared carnocyclin A (bacteriocin IId), sonorensin, and a putative bacteriocin ABC transporter. Such antimicrobial peptide findings indicated that these *E. faecalis* strains had additional competitive advantages against other oral/fang bacteria, which could contribute to their positive selection among the microbiota attempting colonization of *N. nigricollis* venom.

To ascertain if genes unique to these strains were responsible for adaptation to venom, we undertook closer examinations of their draft genome assemblies. Group A isolates shared ~2.9-Mb genomes (2,772 to 2,836 genes), with group B genomes ranging from 3.04 to 3.24 Mb in size (3,128 to 3,282 genes) (Table S3). Including the strains OG1RF and V583, an *E. faecalis* pangenome consisting of 5,130 genes was derived, 1,977 of which formed a core genome that retained venom isolate groupings distinct to OG1RF and V583 (Fig. 4B). After annotation, 235 genes were identified as specific to group A and 321 genes as specific to group B. Most interestingly, among a set of 45 genes unique to both groups, UniProt functional annotation indicated known functions for 46.7% of these (Table S5), with a subset of 15 genes (35.7%) associated with cell wall/membrane integrity and an additional 4-gene set (9.53%) associated with pathogenic foreign protein and toxin defense. Significantly, pathway analysis via DAVID using *Bacillus subtilis* orthologues identified 8-fold enrichment ($P = 0.018$) in the two-component system pathway, specifically genes responsive to cationic antimicrobial peptides, cell wall active antimicrobials, and bacitracin efflux—mechanisms that are collectively compensatory to the well-established antimicrobial activity of venoms.

To ascertain the origin of these isolates and expand our search for proteins with functions opposing venom antimicrobial activity, we extended comparisons to 723 additional *E. faecalis* genome sequences in GenBank. This increased the pangenome by 5-fold (26,412 genes), with only 342 genes highly conserved among all 734 strains. A maximum likelihood tree from the core genome alignment separated the venom-derived strains into different groups (Fig. 4C): placed within distinct subclades relative to isolates of diverse origin globally, including animal, environmental, and human sources, there was no obvious geographical or other link to the two venom-derived strains (Fig. S7). Genome comparisons (70% identity threshold) in this wider context identified 42 genes unique to group A and 97 genes unique to group B (Table S6): among proteins with annotated functions, genes with similar functions were also observed in other *E. faecalis* strains (Tables S7 and S8) such as S3_02356 encoding a colanic acid biosynthesis protein, functionally represented by gene ef95_02851 in strain 7330112-3. The relevance of such genes in adaptation or tolerance to venom will require further molecular characterization in the future.

We next looked for genes whose functions could counteract *N. nigricollis* venom components, such as phospholipase A2 (PLA2), L-amino oxidase, and three-finger peptides (PDB ID 3FTX) that facilitate disruption of membrane integrity in bacteria (25–29). Putative candidates included homologs to acyltransferase-acyl carrier protein synthetase (*Aas*), which was previously reported to protect bacterial cell envelope from human PLA2 in Gram-negative bacteria (30), and homologs to cell wall (*dltA*) and cell membrane (*mprF*) polyanions previously associated with sensitivity to human PLA2 in

**FIG 4** Legend (Continued)

envenomation 1 (E1), and envenomation 2 (E2) samples from three *N. nigricollis* individuals reinforce the two sequence type groupings and highlight within-animal variation (green arcs) indicative of sample-specific variation (lack of reads) across E2 samples in animals 1 and 2. The central plot for each plasmid and animal reflects GC content. All data are represented in 50-nt blocks. (B) Blind maximum likelihood tree of the core genomic alignments for the 6 *N. nigricollis*-derived *E. faecalis* isolates against the V583 and OG1RF reference strains, with color coding referring to the origin of the isolates: light blue, animal 1; dark blue, animal 2; yellow, animal 3. (C) Maximum likelihood tree from concatenated nucleotide sequence alignment of 865 core genes (381,319 bp) from 734 genomes after removing the sites with gaps. The best-fit GTR+I+G4 substitution model was used with 100,00 ultrafast bootstraps and SH-aLRT tests. The tree was rerooted on the longest branch, and branch lengths of <0.001 were collapsed. The scale bar shows number of nucleotide substitutions per site. Branches in red, blue, purple, and green show group A, group B, and clades containing strains V853 and OG1RF, respectively.

*Staphylococcus aureus* (31). Interestingly, the *mprF* gene appeared disrupted into two smaller proteins in one group B *E. faecalis* isolate (V31_01061 and V31_01062) (Table S9). Three copies of sortase family proteins were also detected: sortase A has been associated with the resistance to human PLA2 in *Streptococcus pyogenes* (32). A further 21 genes involved in oxidative stress response and encoding various antioxidative enzymes, such as superoxide dismutase, catalase, and those associated with glutathione metabolism, which would counteract reactive oxygen species produced by venom L-amino acid oxidase activity (25), were also found: these proteins also contribute to the virulence of *E. faecalis* strains (Table S9) (33–35).

Taken together, genome analyses of these *E. faecalis* strains indicated they were well equipped to survive the stress imposed by *N. nigricollis* venom, a tenet supported experimentally, since V583 growth was dose-dependently inhibited at a minimum concentration of 11.7 mg/mL (95% CI, 9.36 to 14.6 mg/mL) and noninhibitory concentration of 2.78 mg/mL (95% CI, 2.21 to 3.50 mg/mL) of filter-sterilized, freeze-dried *N. nigricollis* venom in brain heart infusion broth. In stark contrast, all the venom-derived strains exhibited <30% growth inhibition, even at concentrations of freeze-dried venom of 50 mg/mL (i.e., ∼4× lower than the 208 mg/mL concentration of fresh *N. nigricollis* venom), resulting in ambiguous noninhibitory concentration ranges of 25.2 to 44.0 mg/mL (no CIs calculable), with the group A strain V33 exhibiting no susceptibility to the inhibitory effects of venom (Fig. S8).

**Projecting primary infection clinical risk from venom-tolerant *E. faecalis* isolates.** Since these viable *E. faecalis* strains could potentially infect envenomation wounds, we next looked for known resistance determinants that might facilitate opportunistic primary infection. None of these strains had any acquired resistance genes to any antibiotic classes (Table S10), although all isolates featured *lsaA* (Fig. S9), an intrinsic streptogramin resistance gene (36): Since horizontally acquired genes are largely responsible for resistance to vancomycin, aminoglycosides, macrolides, and tetracycline in *Enterococcus* (37), these strains were considered susceptible to drugs in each of these drug classes. In addition, the absence of known resistance-associated mutations in *gyrA* (DNA gyrase), *parC* (DNA topoisomerase), and the 23S rRNA genes also increased the likelihood of susceptibility to oxazolidinones and fluoroquinolones. Thus, several available antimicrobials would likely be effective in treating infections caused by these strains of *E. faecalis*. However, a gene related to macrolide export (*macB5*) was detected in the venom-tolerant strains (Table S5), and several virulence genes (38, 39) were also identified. Among all isolates, these included conjugative plasmid transfer pheromone genes associated with virulence (*cCF10*, *cOB1*, *cpd*, *cad*, *camE*), endocarditis and biofilm formation-associated pilus subunit genes (*ebpA*, *ebpB*, *ebpC*, *srt*, *pil*), a biofilm on plastic operon gene (*bop*), quorum sensing (*fsrA*, *fsrB*, *fsrC*) and virulence-associated Fsr locus genes, such as gelatinase (*gelE*) and serine protease E (*sprE*), and other important virulence genes, such as those coding for hyaluronidases A and B (*hylA*, *hylB*), thiol peroxidase for oxidative stress resistance (*tpx*), adhesin to collagen of *E. faecalis* (*ace*), endocarditis-specific antigen/*E. faecalis* antigen A (*efaA*), and enterococcal leucine-rich protein A (*elrA*), which prevents macrophage chemotaxis to *E. faecalis* and endocytosis. Furthermore, all six group B isolates had the aggregation substance (*agg*) gene, which enhances macrophage escape by suppressing respiratory burst. Therefore, despite the absence of cytolysin (e.g., *cylA*, *cylB*, *cylM*) and glycopeptide (e.g., *vanA*, *vanC*) resistance genes, both sequence types appeared well equipped to establish infections in human patients following envenomation.

## DISCUSSION

In contrast to the generally held view that venoms are both antimicrobial (1–4) and sterile (6, 7, 40), despite contrasting reports since the 1940s (41), we show that microorganisms can viably colonize venoms of both vertebrates and invertebrates. Moreover, significant adaptation appears to be necessary in genes that counter the mechanisms of action of known, venom-derived antimicrobial peptides and enzymes (4, 5) to attain resistance to venom. Although we documented this in multiple isolates of two independently acquired

*E. faecalis* strains, adaptation appeared to occur in parallel within each of the 3 black-necked cobras from whence the isolates were obtained. It is unclear to what extent this form of parallel convergent evolution extends beyond *N. nigricollis* and *Enterococcus* spp. or other antimicrobial resistance mechanisms, such as antibiotic resistance genes found on mobile genetic elements and working against last-resort antibiotics (42, 43). This work therefore adds to the body of evidence (44) supporting further scrutiny of host-microbe interactions in the venomous microenvironment in understanding microbial adaptation mechanisms to extreme environmental challenges.

Identification of *E. faecalis* as the most prevalent culturable microbe across our European *N. nigricollis* venom samples strikingly reflects three independent clinical reports across Africa and Asia that this nonsporulating microbe is the most common Gram-positive infection cultured from infected envenomation wounds (8–10). Likewise, *E. faecalis* isolates were the most common aerobic Gram-positive isolates in *N. naja* oral swabs (n = 6) (45). Postenvenomation cellulitis and dermatitis, presumed bacterial in nature, were additionally observed in 25% of a 16-case series of *Steatoda nobilis* (false widow spider) envenomations in the United Kingdom and Ireland: one of these required intravenous penicillin and flucloxacillin treatment after hospital admission (46). *S. nobilis* chelicerae were previously found to harbor 11 bacterial taxa and 22 separate bacterial species, including class 2 pathogens; 3 of these 22 species showed multidrug resistance (47). Although explicit genomic evidence connecting venom microbes to envenomation infection remains elusive, in an experimental rabbit model of dermonecrosis (48) caused by *Loxosceles intermedia* (recluse spider) venom, *Clostridium perfringens* recovered from the spider fang and venom enhanced disease symptoms. *Stenotrophomonas*-like bacteria were also found to dominate cone snail venom microbiomes (49), indicating that microbial venom adaptation may extend well beyond snakes, spiders, scorpions, and snails. Furthermore, building upon the few instances of polymicrobial infection reported clinically (8–10), the reports on *L. intermedia* (48) and *Conus* (49) and herein suggest that diverse microbes effectively cocolonize venom glands in a host-species-specific manner, and thus envenomation wounds. Envenomation wound management (40) should therefore extend beyond simply managing the severe tissue damage and necrosis that might be caused by venomous bites to include clinical microbiology on envenomation wounds upon presentation. This would be particularly relevant to individuals immunocompromised through disease or malnutrition (e.g., in developing nations, where envenomation incidence rates are high) or to children on the basis of venom/CFU dose by body weight.

Yet, common microbial diagnostic methods relevant to resource-limited settings have mistaken *E. faecalis* for *Staphylococcus*, which could lead to unfavorable clinical decision making. Although many of the same antibiotics, including vancomycin and linezolid, would be considered for treatment of both staphylococci and enterococci, there are some potential differences. For instance, oxacillin is often employed as a first-line agent to treat *Staphylococcus* (50). This drug is not effective against enterococci, as the use of penicillins for *E. faecalis* infections would typically involve ampicillin, usually in combination with an aminoglycoside (51). In addition, cephalosporins such as cefotaxime are considered second-line therapies for coagulase-negative staphylococci such as *Staphylococcus epidermidis* (52). However, enterococci are intrinsically resistant to this class of drugs, and their prevalence in the gut tends to increase in response to cephalosporin therapy (53). Thus, while the *E. faecalis* strains in this study did not have any known acquired resistance determinants, if they were to cause infections, ensuring their proper identification would be critical to issuing correct treatment and achieving positive clinical outcomes.

It is unclear at present how frequent misidentification events might be. At least one retrospective study reported higher incidence of *Staphylococcus* spp. in envenomation wounds (12), and Blaylock's seminal snake oral microbiota studies also reported *Proteus* and *Staphylococcus* (14): both studies relied on the same methods we used in this study and which misidentified the pathogen. Our results therefore further support use of PCR/sequencing methods as they become more relevant to resource-limited settings (54) and suited to the point of need (55), in line with World Health Organization

ASSURED criteria. Understanding the sensitivity of these methods will be crucial in their reliable implementation in envenomation care. It is therefore noteworthy that despite the limited biomass levels in these samples, species-level OTU analysis on MG-RAST (56) correctly identified *E. faecalis* as one of the principle aerobic isolates in *N. nigricollis* venom. Thus, a simple phylogenetic or metagenomic approach, combined with local herpetogeography knowledge, could quickly and accurately inform clinical action regarding antivenom administration without relying on descriptions or capture of the offending animal or unreliable antibody-based venom identification kits (57).

To conclude, we provide evidence that vertebrate and invertebrate animal venoms host diverse, viable microbiota, with isolates genetically adapted to venom antimicrobials of medical interest against MDR. These results challenge perceptions on the sterility of venom and absence of primary infection risk upon envenomation, pointing to modern nucleic acid technologies to better inform envenomation care and antibiotic use.

## MATERIALS AND METHODS

**Animals and sampling.** All samples analyzed in this study were provided by Venomtech, Ltd., with the exception of freeze-dried *B. arietans* venom (Latoxan, Portes les Valence, France) and air-dried field-collected samples of *B. arietans* venom collected in South Africa (Table 1). Briefly, captive animals were housed in 2-m by 1-m wooden, glass-fronted vivaria with a large hide, thermal gradient, and water *ad libitum*. All procedures for venom collection and swabbing were approved as unregulated under the Animals (Scientific Procedures) Act 1976. Venom was collected by standard techniques; briefly snakes were restrained behind the head and presented to a collection vessel. Snakes freely bit into the vessel until envenomation was observed. Each snake was presented to two sterile collection vessels in succession: one for the first envenomation with potential fang plug and the other for the second flow (labeled E1 and E2, respectively). While the snake was positioned over the second vessel, the oral cavity was swabbed with a sterile swab with individual collection tubes (an invasive sterile swab with transport medium) (DeltaLab; VWR, Lutterworth, United Kingdom). The venom collection vessels were clear, sterile, 125 mL polypropylene containers (Thermo Fisher Scientific, Ltd., Paisley, United Kingdom) covered by 2-by 9 cm² pieces of Parafilm stretched to fit (Thermo Fisher Scientific, Ltd.). The collection vessel was secured to a bench during collection. After collection, aseptically dispensed aliquots were stored in individual 1 mL sterile, DNA-free, polypropylene collection tubes (FluidX, Ltd., Nether Alderley, United Kingdom), at −80°C. Samples collected in the field were from wild puff adders sampled as part of a previous phylogeographic study (58). Venom samples were collected using a method similar to that described for captive animals, except that the entire venom sample was collected in a single collection vessel. Samples were lyophilized by storing <100 $\mu$L venom aliquots in a vacuum-sealed container that was half-filled with silica gel for 1 to 2 days at room temperature. Following drying, venom samples were stored in a refrigerator at 5°C. Snake venom composition and its constituent proteins have been shown to be remarkably resistant to alterations in storage conditions (59) and to degradation during long-term storage (60). We have further observed that our air-dried, field-collected samples for rattlesnake venoms show proteomic profiles identical to those obtained from other research groups using freeze-dried venoms (61, 62). We thus reasoned that variation in storage conditions in this study would be unlikely to have substantially altered the basic properties of the venom substrate available for microbial growth.

*Lasiodora parahybana* and *Poecilotheria regalis* were housed in 5- and 8-L polypropylene boxes (Really Useful Products, Ltd., Normanton, United Kingdom), respectively, with moist vermiculite (Peregrine Livefoods, Ltd., Ongar, United Kingdom), a plastic hide, and a 5 cm water bowl. Arachnids were anaesthetized with a rising concentration of carbon dioxide, the fangs were swabbed with a sterile swab, which was then placed in an individual 1 mL sterile, DNA-free, polypropylene collection tube (FluidX, Ltd.), and venom was subsequently collected from arachnids by electrical stimulation. All samples were stored at −80°C. The same transport swabs (VWR) as those used for snakes were also used for invertebrate oral/fang swabbing. Samples were stored at 4°C and cultured within 24 h of collection.

**Microbial culture.** Aerobic microbial viability was determined by plating swabs or aliquoting 10 $\mu$L volumes of venom samples onto plates containing oxalated whole-horse-blood agar, MacConkey agar, or mannitol salt agar (Thermo Fisher Scientific) and incubating them at 30°C for 72 h. Biochemical isolate identification was undertaken using API strips (20E, 20NE, and Staph), interpreted via the APIWEB interface (bioMérieux, Basingstoke, United Kingdom). All isolates were stored on beads at −80°C at the University of Westminster microbial isolate library. *N. nigricollis* subculture was performed by restoring cryogenically stored bacteria on lysogeny broth agar (Thermo Fisher Scientific), which were grown for 48 h at 30°C, followed by single-colony overnight culture in lysogeny broth (Thermo Fisher Scientific) using aerated culture (300 rpm). MICs and noninhibitory concentrations were determined by broth microdilution assays (63) in brain heart infusion medium by measuring absorbance at the optical density at 600 nm ($OD_{600}$) on a Tecan Spark Cyto 96 plate reader (Tecan, Männedorf, Switzerland) and computed in GraphPad Prism v.9.2.0 according to Lambert and Pearson (64). All bacterial agar and broth materials were purchased from Formedium, Ltd. (Norfolk, United Kingdom).

**DNA extraction.** Neat venom samples or samples diluted in 18 MΩ water previously confirmed as bacterial DNA free by 16S PCR were subjected to DNA extraction using TRIzol, PureLink genomic DNA

kits, or MagMAX cell-free DNA kits (Thermo Fisher Scientific) according to the manufacturer's instructions. For combined extraction of Gram-positive and Gram-negative bacteria from liquid samples, diluted samples were split into equal volumes and processed according to the manufacturer's Gram wall-specific lysis protocols, with lysates combined prior to DNA binding onto columns by simple admixture. DNA content was then analyzed by Nanodrop (Thermo Fisher Scientific) spectrophotometry, and purified material was stored at −80°C until further analysis.

**16S phylogenetic library preparation and sequencing.** For short amplicon library preparation, the hypervariable V3 region of the 16S rRNA gene was amplified from 20 ng of DNA using the primers 5′-CCTACGGGAGGCAGCAG-3′ and 5′-ATTACCGCGGCTGCTGG-3′ (Integrated DNA Technologies BVBA, Leuven, Belgium) (18), 1 U Platinum PCR SuperMix, high fidelity (Thermo Fisher Scientific), and 10 $\mu$M primer mix. The reaction mixtures were incubated at 94°C for 5 min, followed by 30 cycles of 30 s at 94°C, 30 s at 55°C, and 1 min at 72°C and then a final elongation at 72°C for 10 min using a Techne Prime thermal cycler (ColePalmer, Staffordshire, United Kingdom). PCR products (193 bp) were confirmed by 2% (wt/vol) agarose gel electrophoresis in TAE (Tris-acetate-EDTA) buffer (Thermo Fisher Scientific).

Next-generation sequencing (NGS) library preparation was carried out using the Ion Plus fragment library kit according to the manufacturer's instructions (Rev. 3; Thermo Fisher Scientific), except that reaction mixtures were reduced to 1/5 volumes. Pooled libraries were diluted to ∼26 pM for templating on the Ion OneTouch 2 system (Thermo Fisher Scientific) using the Ion PGM Template OT2 200 v.2 kit according to the manufacturer's instructions (Rev. B; Thermo Fisher Scientific). Templated samples were sequenced on the Ion Torrent Personal Genome Machine (PGM) (Thermo Fisher Scientific) system on a single Ion 318 Chip (ThermoFisher Scientific) using the Ion PGM 200 sequencing kit according to the manufacturer's instructions (Rev G.; Thermo Fisher Scientific).

**Whole-genome sequencing.** DNA extracted from cultured isolates was mechanically sheared using the Covaris S220 focused ultrasonicator (Covaris, Brighton, United Kingdom). NGS libraries were generated using the NEBNext fast DNA library prep set for Ion Torrent (New England Biolabs, Hitchin, United Kingdom). Pooled samples were size selected with the LabChip XT (LabChip XT DNA 300 assay kit; PerkinElmer, Seer Green, United Kingdom) and diluted to 26 pM for templating with the Ion OneTouch 2 system using the Ion PGM Template OT2 200 kit. Templated samples were sequenced on the Ion PGM using the Ion PGM Sequencing 200 v.2 reagent kit (Thermo Fisher Scientific) and Ion 318 v.2 Ion Chip (Thermo Fisher Scientific).

**Bioinformatic analyses.** Raw Ion Torrent sequencing data reads were quality controlled and demultiplexed using the standard Ion Server v.4.0 pipeline (Thermo Fisher Scientific). Referenced and *de novo* assemblies were carried out using TMAP v.4.0 and the SPAdes plugin in the Ion Server. Phylogenetic data analyses were carried out after independent data deposition and curation on the MG-RAST v.3.0 pipeline (56) (project IDs MGP5177 and MGP5617), which uses a BLAST approach, and the EBI-METAGENOMICS v.1 pipeline (65) (project ID ERP004004), which uses a hidden Markov model approach. Raw 16S sequencing reads were deposited in the European Nucleotide Archive (accession no. PRJEB4693). Quality control for both resources included length and quality filtering followed by a dereplication step in which sequences with identical 50 nucleotides (nt) in 5′ positions were clustered together. MG-RAST taxonomy annotation involved RNA identification using VSearch, and assignments using a custom database generated by 90% identity clustering of the SILVA, GreenGenes, and RDP prokaryotic databases. EBI-METAGENOMICS identified rRNA using hidden Markov models present in the RDP databases and assigned taxonomy using QIIME and the GreenGenes prokaryotic database.

For postprocessing analyses, the EBI-curated data set was analyzed using MEGAN v.5.5.3 (66). Classical multilocus sequence typing (http://efaecalis.mlst.net/) and cgMLST (21, 22) were carried out using Ridom SeqSphere+ v.4.0 running on a 2-core, 10-GB RAM, 500-GB hard disk Biolinux v.8.0 installation on a VirtualBox virtual machine instance on a 16-GB RAM, 1-TB hard disk Apple iMac. Extended cgMLST analysis to include partially detected loci excluded loci annotated as "failed" due to sequencing error, suggesting genuine *E. faecalis* genomic divergence occurring within each animal. Plasmid detection was carried out using the PlasmidFinder v.1.3 server (67), followed by NCBI BLASTn analysis to detect shorter fragments (e.g., the same 398-nt fragment of *repA-2* in animal 3 isolates (<40% of the full-length gene) at 90.1% identity to the plasmid-borne reference sequence). Single-gene comparisons and multiple-sequence analyses were carried out using TCoffe and MView on the EMBL-EBI server, with base conservation visualized by BoxShade v.3.3.1 on mobyle.pasteur.fr. Genome-level plasmid coverage analyses were carried out by NCBI BLASTn, and comparisons were visualized using Circos v.0.69-4.

The sequencing reads were assembled using SPAdes v.3.9.0 (68), and the draft assemblies were annotated using Prokka (69) before NCBI deposition (BioProject accession no. PRJNA415175). The genome sequences of *E. faecalis* strains V583 and OG1RF (accession no. NC_004668.1 and NC_017316.1, respectively) as well as 723 other *E. faecalis* strains were obtained from GenBank and were reannotated using Prokka to have an equivalence of annotation for comparative analyses. The genomes were compared using the program Roary, with a protein similarity threshold of 70% (70, 71). A maximum likelihood tree was constructed from the core genomic alignment using IQ-Tree (72), with 100,000 ultrafast bootstraps and 100,000 SH-aLRT tests. The tree was visualized using Interactive Tree Of Life (iTOL) (73).

To identify acquired resistance genes, nucleotide BLAST analysis was performed on the ResFinder (74) and NCBI (https://www.ncbi.nlm.nih.gov/pathogens/) resistance gene databases, using cutoffs of 50% length and 85% identity to known resistance determinants. Additional BLAST analysis was performed to identify single nucleotide polymorphisms in the quinolone resistance-determining region (QRDR) of *gyrA* and *parC* (75). Additional mutational analysis was performed on region V of the 23S rRNA-encoding genes (76). Virulence genes were identified using a combination of VirulenceFinder 2.0

(https://cge.cbs.dtu.dk/services/VirulenceFinder/), with default parameters of 60% length and 90% identity to known *Enterococcus* virulence genes (77–79), and manual BLAST, with an E value cutoff of $10^{-5}$. Bacteriocins and ribosomally synthesized and posttranslationally modified peptides were mined using BAGEL4 (http://bagel4.molgenrug.nl) (80).

BLASTp was performed in Ensembl Bacteria (release 38), against *E. faecalis* V583 and *E. faecalis* (GCA_000763645) to obtain further gene IDs from significant matches. *Bacillus subtilis* orthologue gene IDs were collated as this species is the closest relative to *E. faecalis* (VetBact.org), with the most comprehensive genome annotation required for Gene Ontology and KEGG pathway analysis. From the 45 genes unique to venom isolates, useable *B. subtilis* gene IDs were obtained for 20, of which 18 of these were successfully converted to ENTREZ gene IDs using the functional annotation tool (DAVID Bioinformatics resource 6.8) (81, 82), selecting *B. subtilis* as the background species.

**Data availability.** Phylogenetic data were deposited in the MG-RAST (project IDs MGP5177 and MGP5617) and EBI-METAGENOMICS servers (project ID ERP004004). Raw 16S sequencing reads were deposited in the European Nucleotide Archive (accession no. PRJEB4693). Annotated draft *E. faecalis* genome assemblies were deposited in NCBI (BioProject accession no. PRJNA415175).

## SUPPLEMENTAL MATERIAL

Supplemental material is available online only.
**SUPPLEMENTAL FILE 1**, PDF file, 2.4 MB.
**SUPPLEMENTAL FILE 2**, XLSX file, 1 MB.

## ACKNOWLEDGMENTS

We thank Pamela Greenwell and Caroline Smith for invaluable input on nonstandard DNA extraction methodology options suited to unusual samples, Patrick Kimmit for input on microbial characterization, and Peter Gibbens for housing and venom collection from captive *N. nigricollis* and *B. arietans*.

This work was funded by the University of Westminster, University of Northumbria, and Venomtech, Ltd. The views expressed in this article are those of the authors and do not necessarily reflect the official policy of the Department of Health and Human Services, the U.S. Food and Drug Administration, or the U.S. Government.

M.M.G.L. and T.D.L. sampled and C.T. and S.T. prepared the library of captive animal venoms. W.W. and A.B. collected and prepared the wild snake samples. E.E., J.D.T., and S.A.M. optimized and performed the DNA extractions and 16S PCR. J.D.T. and E.E. performed the preliminary and main study library preps and next-generation sequencing experiments, respectively. A.D., H.D., L.A.S.S., and S.A.M. performed the phylogenetic data quality control, curation, and analysis. K.F.R. and S.A.M. performed the microbial cultures and biochemical characterization. M.K.-V. and L.U. grew the *E. faecalis* isolates and performed the whole-genome sequencing. M.K.-V., K.W., and S.A.M. performed the *E. faecalis* isolate genomic characterization and MLST+ analysis. G.H.T. performed *E. faecalis* resistome analysis. V.S. performed the *E. faecalis* isolate pangenome data reduction, and S.T. identified the venom resistance Gene Ontology subset. G.H.T., V.S., and S.A.M. identified the virulence genes, and S.A.M. identified the bacteriocin content in *E. faecalis* isolates. S.A.M. conceived the study and designed experiments with S.T. S.D. performed the MIC/NIC curve assays. All authors contributed equally to the overall interpretation of the data set and manuscript preparation.

We declare no conflict of interest.

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
