## [Reviewer comments · Microbiology Spectrum]

Microbiology Spectrum

Bacterial adaptation to venom in snakes and arachnida

Elham Esmailshirazifard, Louise Usher, Carol Trim, Hubert Denise, Vartul Sangal, Gregory Tyson, Axel Barlow, Keith Redway, Joe Taylor, Myrto Kremmyda-Vlachou, Sam Davies, Tessa Loftus, Mikaella Lock, Katie Wright, Andrew Dalby, Lori Snyder, Wolfgang Wuster, Steve Trim, and Sterghios Moschos

Corresponding Author(s): Sterghios Moschos, Northumbria University

Review Timeline:

Submission Date:	December 1, 2021
Editorial Decision:	February 6, 2022
Revision Received:	April 6, 2022
Editorial Decision:	April 13, 2022
Revision Received:	April 13, 2022
Accepted:	April 14, 2022

Editor: Olaya Rendueles Garcia

Reviewer(s): Disclosure of reviewer identity is with reference to reviewer comments included in decision letter(s). The following individuals involved in review of your submission have agreed to reveal their identity: Bruno Amorim-Carmo (Reviewer #1); Sylvie REBUFFAT (Reviewer #3)

Transaction Report:

DOI: <https://doi.org/10.1128/spectrum.02408-21>

February 6, 2022

Dr. Sterghios A Moschos
Northumbria University
Applied Sciences
Ellison building
Ellison Place
Newcastle Upon Tyne, Tyne and Wear NE1 8ST
United Kingdom

Re: Spectrum02408-21 (Microbial adaptation to venom in snakes and spiders)

Dear Dr. Sterghios A Moschos:

Thank you for submitting your manuscript to Microbiology Spectrum.

First, I'd like to apologize for the delay in submitting my decision. It has been difficult to secure enough reviewers for the manuscript.

It has been reviewed by three different experts in the field. They found your experiments clear and well-performed, and they consider the study of interest. However, they also raise some issues that must be attended to. More particularly, reviewers #2 and #3 point out that some sections could be more focused and that the manuscript should be better streamlined, and shortened. In particular, it could benefit from better presentation of the aims of the study. I thus invite you to revise the manuscript style and presentation in a significant way, in order to make it more clear and easy-to follow.

When submitting the revised version of your paper, number the lines of your manuscript and please provide (1) point-by-point responses to the issues raised by the reviewers as file type "Response to Reviewers," not in your cover letter, and (2) a PDF file that indicates the changes from the original submission (by highlighting or underlining the changes) as file type "Marked Up Manuscript - For Review Only". Please use this link to submit your revised manuscript - we strongly recommend that you submit your paper within the next 60 days or reach out to me. Detailed instructions on submitting your revised paper are below.

Link Not Available

Sincerely,

Olaya Rendueles Garcia

Journals Department
Reviewer comments:

Reviewer #1 (Comments for the Author):

Adding numbering to the lines of the manuscript makes correction easier.

Reviewer #2 (Comments for the Author):

Overall, the identification of *Enterococcus* species resistant to venom is interesting but the presentation of the data in this manuscript makes the purpose of this study difficult to understand. It almost feels that there are two different stories - one about the microbiome and one about the characterisation of the *Enterococcus* strain. At the moment there are no clear aims and the manuscript is confusing to read with lots of data in supplements and no clear path to understand what the manuscript is about. The presentation is very long and the results are described in a largely unnecessary way and should be shortened - in this regard, it almost felt that the results were combined with a discussion, but then there was another discussion on top. The discussion itself does not nicely wrap up the story and contributes to further distraction as e.g., clinical implications that are not necessarily relevant to this study were discussed.

Another major limitation about this study and the overall idea of microbes in venom is that the microbial adaptation to venom was only demonstrated in two *E. faecalis* isolates from one snake species - so it is not clear how common microbial adaptation to venom would be in multiple venomous species. The focus is almost exclusively on *N. nigricollis*, so this should be reflected in the manuscript - at the moment, this study, as presented, shows preliminary evidence of their idea, but requires follow up experiments to show whether adaptation is a common phenomenon.

Other comments:

- Two scorpion species are mentioned but not followed up - it is unclear why they were not discussed?
- There are numerous grammatical errors - too many to mention; the references need to be formatted and updated.

Reviewer #3 (Comments for the Author):

The paper from E. Esmaeilshirazifard et al. describes the identification of venom tolerant bacteria in snakes and spiders and their adaptation to the venom conditions. It shows that genes encoding proteins involved in different strategies to resist to antimicrobial compounds present in the venoms are acquired to drive this adaptation.

The authors examined the compositions of the oral and venom microbiota of five snake, two spider and two scorpion species. They studied more deeply the *Naja nigricollis* venom, which was shown to host two novel *Enterococcus faecalis* strains resistant to the venom conditions, due for a part to improved membrane stability and integrity. Among other most important results the authors showed the absence of correlation between the bacterial species found in the venom and the animal class or the taxon, but rather with anatomical or physiological constraints, that the venom-associated bacteria are specific to the venom environment, and that plasmidic mobile elements, especially those that encode bacteriocins contribute to the genomic divergence of *E. faecalis* within each animal.

This in-depth study of the bacterial content of venoms from animals belonging to two different classes, Reptilia and Arachnida, questions the assumption that venoms are sterile sources of antimicrobial compounds. It affords well established conclusions that should be explored further. The data presented were obtained from a logical design and in a rigorous fashion. The schemes and figures are clear and the manuscript is well written. However, a number of points detailed below need clarification.

Specific comments

- 1- The title of the paper "Microbial adaptation to venom in snakes and spiders" suggests that different microbes are considered while the paper only describes bacteria. Moreover, not only spiders but snakes and scorpions are studied. Thus the title could be changed to "Bacterial adaptation to venom in snakes and arachnida".
- 2- The term "microbiome" is used extensively throughout the paper without differentiating "microbiome" from "microbiota". These two terms have been clearly defined considering on one side strictly the microbial communities that occupy a given habitat that are defined as the "microbiota", or on the other side the whole system that is defined as "microbiome", including both the microbial communities and their structural elements (nucleic acids, proteins, polysaccharides,...) and metabolites involved in the occupation of a niche (see Berg G. et al 2020, *Microbiome*, 8: 103). The authors have to check the manuscript for the undifferentiated use of "microbiome" and change it to "microbiota" when required. Moreover, the use of the term 'flora' (page 4 line 17, p. 6 line 16, p. 16, line 18) to name the microbial communities in various compartments, is currently out of date and the term "microbiota" should be preferred. Changes have to be made appropriately. For instance page 4 line 17 the title "The snake venom microbiome varies on account of host species and not on account of the oral flora" should be "The snake venom microbiota varies on account of host species and not on account of the oral microbiota", or page 5, lines 16, 17, change "flora" and "microbiome" to "microbiota".
- 3- The authors have to specify in the paper that they identified all bacterial species, cultivable and non-cultivable, in venoms or

oral cavities but that only cultivable bacteria have been considered ("aerobic viability") to assess adaptation to the venom environment.

4- The paper is well written, but some sections, especially section "Pangenomic evidence of *E. faecalis* isolate adaptation to venom" and the discussion part could be more concise or more focused.

5- Materials and methods:

- it is unclear (page 18) how snake venom samples were conserved before lyophilization (-80°C as for spiders?): please specify. This is especially questionable for field-collected samples. Moreover it is suggested line 22 page 6 that venom samples collected from wild animals could have been only air-dried and not conserved at very low temperature and further lyophilized, which very presumably should impact both the venom chemical content and therefore the bacterial strains maintained in this medium. This has to be absolutely clarified and justified.

- it is unclear page 17 lines 18-19 and when comparing to the legend to figures, which samples analyzed are originating from Venomtech and Latoxan and which have been collected from captive or wild animals. Furthermore, it is unclear when commercial lyophilized samples (with unknown caution taken to avoid contamination by external bacteria...) and collected samples were used. This has to be specified very precisely.

- there is no information about treatments of scorpions and their venom in Materials and methods, why they are considered in Table 1 and in the text; please complete.

6- While two scorpions have been included in the study, neither conclusions nor discussion are provided about these venoms and their potential bacterial content. Either data related to scorpion venoms have to be described and discussed in the manuscript, or if they do not afford interesting information, they have to be suppressed from the study.

7- Even if the genes identified in *E. faecalis* and discussed in the paper are trivial and well-known, the functions of the encoded proteins should be specified along the paper to make it more easily understandable by readers that are not strictly involved in the topic, and thus allow broadening the interested scientific community. Otherwise, a list of abbreviations could be provided. In addition or alternatively in certain cases, UniProt codes for the most critical genes identified could be provided.

8- The mammalian origin of strain *E. faecalis* V583 (usually found in soils, waters, food etc and very probably arising from mammalian gastrointestinal tract via feces), which is the closest relative to *E. faecalis* strains from *N. nigricollis* venom has to be specified in the paragraph page 7 lines 11-22.

9- Resistance of the venom-derived *E. faecalis* strains to the venom antimicrobial content is evaluated in the paper, but pathogenicity of the strains is not examined, while one of the objectives of the paper is to demonstrate that infections associated with envenomations can be due to pathogens contained in the venom. *Enterococcus* spp. are opportunistic pathogens and nothing indicates that the strains isolated from venom samples could be commensals or pathogens. Numerous virulence factors have been identified in *Enterococcus* species and especially in *E. faecalis*. Did the authors investigate this aspect by searching for specific genes encoding virulence factors, such as gelatinase (*gelE*, *fsrB*), cytolysin (*cyl*), aggregation substance, etc (see references Nami Y. et al 2015, *Front. Microbiol.* 6:782; Hashem Y.A. et al. 2021, *Infect Drug Res.* 2021:14)? This important information should be added.

10- Furthermore, the authors consider the antimicrobial molecules usually described as present in venoms, including antimicrobial peptides, but do not take into account that the antimicrobial peptides produced by the bacteria (bacteriocins) compete against other closely related strains in the same niche and thus could also contribute to the selection of the *E. faecalis* isolates present in the venom by killing other bacteria genera close to the producer thanks to their bacteriocins. The authors mention briefly the production of bacteriocin 41 (Bac41), but many other bacteriocins have been identified from *Enterococcus* strains (mainly enterocins L50A/L50B, A, B, P, Q, X, AS-48; bacteriocins 32, 43, 51). Did the authors search for the genes encoding such well-known bacteriocins? The information has to be added and the role of the bacteriocins in adaptation of bacteria to venom conditions has to be discussed.

11- Figures:

- Figure 1: Why is B3 absent from Figure 1B, while the legend indicates "The 8 wild (red dots B1-B8)"?

- Figure 3 is unclear: B and D are lacking.

12- Minor points

- Page 5, title last line change "A fifth of the *N. nigricollis* venom microbiome is distinct to that of fangs" to "A fifth of the *N. nigricollis* venom microbiota is distinct from that of fangs"

- Typing mistakes - page 7 line 4, in "*Staphylococcus* spp.", "spp" should not be in italics; page 11: line 9 change "Biotin transporter" to "biotin transporter", "Bacterial regulatory" to "bacterial regulatory", "gntR" to "GntR"; line 15 change "L-Amino acid" to "L-amino acid"; page 12: line 4 change "was" to "were", line 7 change "L-Amino acid" to "L-amino acid", line 9 change "l-amino acid" to "L-amino acid"; page 13 "macB5" should be in italics; page 37, Table 1 footnote: use the greek letter instead of "ul" for microliter; page 38 Table 2: change "iqil" to "iqiL"; Legend to figure 3: line 10, *N. nigricollis* should be in italics.

Staff Comments:

Preparing Revision Guidelines

To submit your modified manuscript, log onto the eJP submission site at <https://spectrum.msubmit.net/cgi-bin/main.plex>. Go to

Author Tasks and click the appropriate manuscript title to begin the revision process. The information that you entered when you first submitted the paper will be displayed. Please update the information as necessary. Here are a few examples of required updates that authors must address:

Please return the manuscript within 60 days; if you cannot complete the modification within this time period, please contact me. If you do not wish to modify the manuscript and prefer to submit it to another journal, please notify me of your decision immediately so that the manuscript may be formally withdrawn from consideration by Microbiology Spectrum.

Minor revisions:

1. Correct the word “L-Amino acids” for L-amino acids throughout the manuscript
2. Correct the measurement units throughout the text. The unit of measure liter (L) must be written with a capital letter in accordance with the International System of Units (SI). For example, on **page 12**, change (ng/ml) to (ng/mL).
3. I believe the correct term for what the authors called "96 well format assays" in the "Microbial culture" section on **page 19** would be "broth microdilution assay" according to CLSI.
4. I could not find the reference by Lambert et al. cited in the manuscript in the section "Microbial culture" on **page 19** to check the methodology used. I do not know if the authors used the CLSI (Clinical & Laboratory Standards Institute) as a reference for microbiological tests. Please add the reference.

Bacterial adaptation to venom in snakes and arachnida.

Response to Reviewers: Point by point changes

Reviewer #1

1. Correct the word "L-Amino acids" for L-amino acids throughout the manuscript

Done

2. Correct the measurement units throughout the text. The unit of measure liter (L) must be written with a capital letter in accordance with the International System of Units (SI). For example, on page 12, change (ng/ml) to (ng/mL).

Done

3. I believe the correct term for what the authors called "96 well format assays" in the "Microbial culture" section on page 19 would be "broth microdilution assay" according to CLSI.

Done

4. I could not find the reference by Lambert et al. cited in the manuscript in the section "Microbial culture" on page 19 to check the methodology used. I do not know if the authors used the CLSI (Clinical & Laboratory Standards Institute) as a reference for microbiological tests. Please add the reference.

We thank the reviewer for noting the omission of the Lambert and Pearson (2000) method which has now been introduced in the manuscript (reference 64) and the relevant section has been extended to indicate MIC and NIC were computed in GraphPad Prizm v9.2.0.

We note that there is no standardised broth microdilution assay protocol specified in the UK Health Security Agency Reference Laboratories Colindale Bacteriology Reference Department user manual, or the English Surveillance Programme for Antimicrobial Utilisation and Resistance which applies in the UK. Rather, a mix of CLSI and EUCAST guidelines are used for NIC and MIC calculation, depending on the antibiotic in question. In our laboratories we comply to CLSI standard operating procedures with respect to all laboratory preparation and maintenance protocols (CLSI M07; now reference 63), but for this new antimicrobial substance we used a well-cited experimental MIC and NIC calculation protocol since no national or international standard applies.

Reviewer #2

Overall, the identification of Enterococcus species resistant to venom is interesting but the presentation of the data in this manuscript makes the purpose of this study difficult to understand. It almost feels that there are two different stories - one about the microbiome and one about the characterisation of the Enterococcus strain. At the moment there are no clear aims and the manuscript is confusing to read with lots of data in supplements and no clear path to understand what the manuscript is about. The presentation is very long and the results are described in a largely

unnecessary way and should be shortened - in this regard, it almost felt that the results were combined with a discussion, but then there was another discussion on top. The discussion itself does not nicely wrap up the story and contributes to further distraction as e.g., clinical implications that are not necessarily relevant to this study were discussed.

We understand and agree with the reviewer's position that this manuscript feels like two stories- one about the microbiome and one about the characterisation of the *Enterococcus faecalis* strains (2). Indeed our work was carried out in two stages- one describing the microbiota, and one understanding the nature (*E. faecalis* instead of *Staphylococcus*) and adaptation of the most abundant viable aerobes in at least one of the host species.

We feel that reporting on the microbiota in venoms in itself is interesting, because it sets the stage with regards to the evolutionary gradient between envenomation apparatus and venom itself, across which adaptation takes place. However, the natural question that arises is "how do these bacteria manage to survive in venom?" Therefore, it is of equal interest to report upon the mechanisms of adaptation to these extreme environments. We would therefore like to thank the reviewer for their position and, at the editors' discretion, we would therefore consider splitting the manuscript into two papers covering each aspect independently.

To make the aims explicit, we have added a new paragraph at the end of the introduction (page 5, lines 15-21).

In line with the editors and reviewer no.2's recommendation, we have shortened the presentation of results to the extent possible without removing any of the existing supplementary datasets, and rephrased the main text to improve upon clarity. In line with reviewer no. 3's request for additional *E. faecalis* genome analysis, however, we have expanded upon virulence findings across the 9 isolates in the results section (page 14 line 1 – 14).

The discussion section therefore now brings together the evidence in the field that bacteria adapt to venoms, to then colonise envenomation wounds opportunistically during the envenomation injury, which is mechanistically substantiated in this paper at least for *E. faecalis* venom isolates.

In consolidating the clinical implications to the discussion section as requested by reviewer no. 2 (page 16, line 11 - 2), and in line with the interests expressed by reviewer no. 3, we now substantiate more readily how biochemical misdiagnosis could affect clinical management, with recommendations on how to overcome these problems. Given the size of morbidity and mortality burden of snakebite it is our view that highlighting these issues is necessary to improve upon management practice.

Another major limitation about this study and the overall idea of microbes in venom is that the microbial adaptation to venom was only demonstrated in two *E. faecalis* isolates from one snake species - so it is not clear how common microbial adaptation to venom would be in multiple venomous species. The focus is almost exclusively on *N. nigricollis*, so this should be reflected in the manuscript - at the moment, this study, as presented, shows preliminary evidence of their idea, but requires follow up experiments to show whether adaptation is a common phenomenon.

We agree with the reviewer and therefore have shortened the relevant discussion section and explicitly state that adaptation is documented in *E. faecalis* isolates, but it remains unclear to which extent this paradigm extends beyond *N. nigricollis* or *E. faecalis* (page 14, lines 19-23). We then call for further work in this field (page 15, lines 4-6)

Two scorpion species are mentioned but not followed up - it is unclear why they were not discussed?

We thank the reviewer for highlighting the limited analysis of this aspect of our work. Accordingly, we have removed the scorpion data from the manuscript.

There are numerous grammatical errors - too many to mention; the references need to be formatted and updated.

We thank the reviewer for highlighting the need for further proofreading and have incorporated all the recommendations from Reviewer no. 3 and had native English speakers from within the author list review grammatical content.

We have now also revised the manuscript references to the Journal style.

Reviewer #3

1- The title of the paper "Microbial adaptation to venom in snakes and spiders" suggests that different microbes are considered while the paper only describes bacteria. Moreover, not only spiders but snakes and scorpions are studied. Thus the title could be changed to "Bacterial adaptation to venom in snakes and arachnida".

We thank the reviewer for their recommendation and have accordingly modified the manuscript title to "Bacterial adaptation to venom in snakes and arachnida".

2- The term "microbiome" is used extensively throughout the paper without differentiating "microbiome" from "microbiota". These two terms have been clearly defined considering on one side strictly the microbial communities that occupy a given habitat that are defined as the "microbiota", or on the other side the whole system that is defined as "microbiome", including both the microbial communities and their structural elements (nucleic acids, proteins, polysaccharides,..) and metabolites involved in the occupation of a niche (see Berg G. et al 2020, Microbiome, 8: 103). The authors have to check the manuscript for the undifferentiated use of "microbiome" and change it to "microbiota" when required. Moreover, the use of the term 'flora' (page 4 line 17, p. 6 line 16, p. 16, line 18) to name the microbial communities in various compartments, is currently out of date and the term "microbiota" should be preferred. Changes have to be made appropriately. For instance page 4 line 17 the title "The snake venom microbiome varies on account of host species and not on account of the oral flora" should be "The snake venom microbiota varies on account of host species and not on account of the oral microbiota", or page 5, lines 16, 17, change "flora" and "microbiome" to "microbiota".

We thank the reviewer for their recommendation and have reviewed and corrected individually all instances where the terms were used incorrectly.

3- The authors have to specify in the paper that they identified all bacterial species, cultivable and non-cultivable, in venoms or oral cavities but that only cultivable bacteria have been considered ("aerobic viability") to assess adaptation to the venom environment.

We thank the reviewer for their recommendation. Accordingly, in the opening sentence of the section concerning viability (page 8, line 5) we explicitly state "*After identifying all cultivable and non-cultivable bacterial species... we next proceeded to examine if cultivable aerobes could be recovered... as an indication of adaptation to venom*". Accordingly, in discussing findings with *B. arietans* we extend statements on the impact of venom handling on microbial viability to "*at least for aerobic bacteria*".

4- The paper is well written, but some sections, especially section "Pangenomic evidence of *E. faecalis* isolate adaptation to venom" and the discussion part could be more concise or more focused.

We thank the reviewer for their recommendation. In line with reviewer no. 2's request and the editor's request, we have reworded this section to reduce its length and for clarity purposes.

We have also moved the analysis on the clinical implications of the *E. faecalis* genomes to the discussion section (page 16, lines 11-2). We have made efforts to reduce the verbosity of the discussion section, focusing now on bringing together the evidence beyond our work pointing to more widespread adaptation of microbes to venoms, and contextualising our findings to the clinical implications of envenomation-derived infections. In line with reviewer no. 3's interests in the virulence of these *E. faecalis* isolates, we feel this section is important given epidemiological evidence and the morbidity and mortality burden of snakebite across resource-limited territories.

5- Materials and methods:

- it is unclear (page 18) how snake venom samples were conserved before lyophilization (-80°C as for spiders?): please specify. This is especially questionable for field-collected samples. Moreover it is suggested line 22 page 6 that venom samples collected from wild animals could have been only air-dried and not conserved at very low temperature and further lyophilized, which very presumably should impact both the venom chemical content and therefore the bacterial strains maintained in this medium. This has to be absolutely clarified and justified.

We thank the reviewer for their highlighting this omission in such a crucial part of the methodological reporting in our manuscript. We have accordingly:

- Expanded Page 18 lines 15-17 to state how snake and spider venoms were preserved between collection and analysis.
- Expanded upon the microbiota results presentation section (page 6, line 22 – page 7 line 2) to explicitly state the unknown provenance of the Latoxan sample, and how

- the air-drying of the wild-caught samples could have been reasonably expected to have a big impact on the 16S profile, but it does not appear to have done so.
- Expanded upon the justification of the handling protocol for venom samples collected in the wild (page 18 line 21 page 19 line 5).

Whilst we agree that the air drying and hyperconcentration of the *B. arietans* wild venoms could have reasonably destroyed any viable bacteria, we note that the Venomtech animal venom did not yield any viable aerobic isolates either. Therefore, in line with the justification stated in the manuscript and the experimental observations presented we feel that the statement restricted to aerobic viability (page 8, line 14) is appropriate given consistency of results.

- it is unclear page 17 lines 18-19 and when comparing to the legend to figures, which samples analyzed are originating from Venomtech and Latoxan and which have been collected from captive or wild animals. Furthermore, it is unclear when commercial lyophilized samples (with unknown caution taken to avoid contamination by external bacteria...) and collected samples were used. This has to be specified very precisely.

We have clarified readability by annotating the relevant entries in Table 1 with L and B1-8, and adding the relevant labels to legend of Figure S3. We have additionally explicitly stated in the methods section (page 18, lines 2-3) which samples were freeze-dried, and which samples were air-dried.

- there is no information about treatments of scorpions and their venom in Materials and methods, why they are considered in Table 1 and in the text; please complete.

We thank the reviewer for highlighting the limited analysis of this aspect of our work. Accordingly, we have removed the scorpion data from the manuscript.

6- While two scorpions have been included in the study, neither conclusions nor discussion are provided about these venoms and their potential bacterial content. Either data related to scorpion venoms have to be described and discussed in the manuscript, or if they do not afford interesting information, they have to be suppressed from the study.

We thank the reviewer for highlighting the limited analysis of this aspect of our work. Accordingly, we have removed the scorpion data from the manuscript.

7- Even if the genes identified in *E. faecalis* and discussed in the paper are trivial and well-known, the functions of the encoded proteins should be specified along the paper to make it more easily understandable by readers that are not strictly involved in the topic, and thus allow broadening the interested scientific community. Otherwise, a list of abbreviations could be provided. In addition or alternatively in certain cases, UniProt codes for the most critical genes identified could be provided.

We thank the reviewer for highlighting the need to expand on the names of the genes discussed (*repA-2*, *gyrA*, *parC*, *macB5*). These, alongside new entries pertaining to the

virulence and bacteriocin genes have now been identified in terms of function (page 10, line 12, 21, 21; page 13, lines 16, 20-23; page 14 lines 1-14)

8- The mammalian origin of strain *E. faecalis* V583 (usually found in soils, waters, food etc and very probably arising from mammalian gastrointestinal tract via feces), which is the closest relative to *E. faecalis* strains from *N. nigricollis* venom has to be specified in the paragraph page 7 lines 11-22.

We thank the reviewer for this recommendation. A statement thereto has been added to the recommended section (page 9 lines 4-5).

9- Resistance of the venom-derived *E. faecalis* strains to the venom antimicrobial content is evaluated in the paper, but pathogenicity of the strains is not examined, while one of the objectives of the paper is to demonstrate that infections associated with envenomations can be due to pathogens contained in the venom. *Enterococcus* spp. are opportunistic pathogens and nothing indicates that the strains isolated from venom samples could be commensals or pathogens. Numerous virulence factors have been identified in *Enterococcus* species and especially in *E. faecalis*. Did the authors investigated this aspect by searching for specific genes encoding virulence factors, such as gelatinase (gelE, fsrB), cytolysin (cyl), aggregation substance, etc (see references Nami Y. et al 2015, Front. Microbiol. 6:782; Hashem Y.A. et al. 2021, Infect Drug Res, 2021:14)? This important information should be added.

We thank the reviewer for this recommendation. Accordingly, we have used VirulenceFinder and manual searching (methods detail added in page 23 lines 23 – page 24 line 4) and findings are presented in the relevant results section, succinctly in line with other requests from both reviewers and without the addition of yet another supplementary table, in page 13 line 23 - page 14 line 14), citing Hashem *et al.* and Nami *et al.*

10- Furthermore, the authors consider the antimicrobial molecules usually described as present in venoms, including antimicrobial peptides, but do not take into account that the antimicrobial peptides produced by the bacteria (bacteriocins) compete against other closely related strains in the same niche and thus could also contribute to the selection of the *E. faecalis* isolates present in the venom by killing other bacteria genera close to the producer thanks to their bacteriocins. The authors mention briefly the production of bacteriocin 41 (Bac41), but many other bacteriocins have been identified from *Enterococcus* strains (mainly enterocins L50A/L50B, A, B, P, Q, X, AS-48; bacteriocins 32, 43, 51). Did the authors searched for the genes encoding such well-known bacteriocins? The information has to be added and the role of the bacteriocins in adaptation of bacteria to venom conditions has to be discussed.

We thank the reviewer for bringing to our attention this aspect of opportunistic dominance for the *E. faecalis* strains identified in *N. nigricollis* venoms in our study. To this end we have used the well cited bacteriocin miner BAGEL4 to identify relevant genes in our isolates and included this information in the results section (page 10, line 19 – page 11 line 3).

11- Figures:

- Figure 1: Why is B3 absent from Figure 1B, while the legend indicates "The 8 wild (red dots B1-B8)"?

We thank the reviewer for spotting this reporting oversight. An explanatory note has been added to Figures 1 and S3 on the ~100x lower read depth obtained from sample B3, leading to its exclusion from these analyses.

- Figure 3 is unclear: B and D are lacking.

Labels B and D have been added to the figure.

12- Minor points

- Page 5, title last line change "A fifth of the *N. nigrocollis* venom microbiome is distinct to that of fangs" to "A fifth of the *N. nigrocollis* venom microbiota is distinct from that of fangs"

- Typing mistakes - page 7 line 4, in "*Staphylococcus spp.*", "*spp.*" should not be in italics; page 11: line 9 change "Biotin transporter" to "biotin transporter", "Bacterial regulatory" to "bacterial regulatory", "gntR" to "GntR"; line 15 change "L-Amino acid" to "L-amino acid"; page 12: line 4 change "was" to "were", line 7 change "L-Amino acid" to "L-amino acid", line 9 change "l-amino acid" to "L-amino acid"; page 13 "*macB5*" should be in italics; page 37, Table 1 footnote: use the greek letter instead of "ul" for microliter; page 38 Table 2: change "iqil" to "iqiL"; Legend to figure 3: line 10, *N. nigricollis* should be in italics.

All changes made as requested.

April 13, 2022

Dr. Sterghios A Moschos
Northumbria University
Applied Sciences
Ellison building
Ellison Place
Newcastle Upon Tyne, Tyne and Wear NE1 8ST
United Kingdom

Re: Spectrum02408-21R1 (Bacterial adaptation to venom in snakes and arachnida)

Dear Dr. Sterghios A Moschos:

Thank you for submitting your manuscript to Microbiology Spectrum. As you will see your paper is very close to acceptance. I appreciate greatly your efforts to comply with the reviewers demands. The manuscript is now more focused and easier to read. there are still some minor text edits that you should implement prior to acceptance (see below).

Please modify the manuscript along the lines I have recommended. As these revisions are quite minor, I expect that you should be able to turn in the revised paper in less than 30 days, if not sooner. If your manuscript was reviewed, you will find the reviewers' comments below.

When submitting the revised version of your paper, please provide (1) point-by-point responses to the issues I raised in your cover letter, and (2) a PDF file that indicates the changes from the original submission (by highlighting or underlining the changes) as file type "Marked Up Manuscript - For Review Only". Please use this link to submit your revised manuscript. Detailed instructions on submitting your revised paper are below.

Link Not Available

Sincerely,

Olaya Rendueles Garcia

Reviewer comments:

Reviewer #3 (Comments for the Author):

The revised version of the paper from E. Esmailshirazifard et al., which affords evidence of the presence of venom tolerant bacteria in snakes and spiders and their adaptation to the venom conditions, has been significantly improved. All concerns raised and specific comments have been addressed appropriately by the authors and without eluding the questions. The manuscript has been modified accordingly. Following my proposition, the title has been modified and is now sound. Moreover, the aims of the paper have been clarified, the paper has been shortened and is now better articulated. A few minor points mentioned below need being corrected.

- 1- The Abstract needs being improved by using a past tense (preterit) rather than the present for describing what has been done or observed: Change "observe" to "observed" line 19 page 2, "evidence" to "evidenced" line 20 page 2, "identify" to "identified" line 2 page 3, and "an additional 45 genes" to "45 additional genes" line 3 page 3.
- 2- Introduction page 5 line 18, change "examine" to "examined" and "employ" to "employed"
- 3- Page 11, line 11 change "group groups" to "groups; line 13, change "an additional 4 genes" to "an additional 4 gene set"
- 4- Page 12, line 6 change "faecallis" to "faecalis"
- 5- Page 13, line 15 change "antimicrobial" to "antibiotic" to better express that conventional antibiotics are considered here.

6- Page 15 line 12 change "was additionally" to "were additionally"

7- List of references:

-Ref 46 and 47: author names are lacking;

-Ref 23: two years are indicated 2003 and (1979): suppress 1979

-Ref 44 is incomplete (what does Toxicon X4 means?)

-Names of bacteria, snakes etc should be in italics; conform to the journal requirements but if capital letters are used for each name in the titles this should be homogeneous over the list; similar, the names of the journals should be abbreviated using the conventional abbreviations and not alternatively full names/abbreviations.

Preparing Revision Guidelines

- point-by-point responses to the issues I raised in your cover letter
- Upload a compare copy of the manuscript (without figures) as a "Marked-Up Manuscript" file.
- Each figure must be uploaded as a separate file, and any multipanel figures must be assembled into one file.
- Manuscript: A .DOC version of the revised manuscript
- Figures: Editable, high-resolution, individual figure files are required at revision, TIFF or EPS files are preferred

Please return the manuscript within 60 days; if you cannot complete the modification within this time period, please contact me. If you do not wish to modify the manuscript and prefer to submit it to another journal, please notify me of your decision immediately so that the manuscript may be formally withdrawn from consideration by Microbiology Spectrum.

Bacterial adaptation to venom in snakes and arachnida.

Response to Reviewers: Point by point changes

Reviewer #1

1. Correct the word "L-Amino acids" for L-amino acids throughout the manuscript

Done

2. Correct the measurement units throughout the text. The unit of measure liter (L) must be written with a capital letter in accordance with the International System of Units (SI). For example, on page 12, change (ng/ml) to (ng/mL).

Done

3. I believe the correct term for what the authors called "96 well format assays" in the "Microbial culture" section on page 19 would be "broth microdilution assay" according to CLSI.

Done

4. I could not find the reference by Lambert et al. cited in the manuscript in the section "Microbial culture" on page 19 to check the methodology used. I do not know if the authors used the CLSI (Clinical & Laboratory Standards Institute) as a reference for microbiological tests. Please add the reference.

We thank the reviewer for noting the omission of the Lambert and Pearson (2000) method which has now been introduced in the manuscript (reference 64) and the relevant section has been extended to indicate MIC and NIC were computed in GraphPad Prizm v9.2.0.

We note that there is no standardised broth microdilution assay protocol specified in the UK Health Security Agency Reference Laboratories Colindale Bacteriology Reference Department user manual, or the English Surveillance Programme for Antimicrobial Utilisation and Resistance which applies in the UK. Rather, a mix of CLSI and EUCAST guidelines are used for NIC and MIC calculation, depending on the antibiotic in question. In our laboratories we comply to CLSI standard operating procedures with respect to all laboratory preparation and maintenance protocols (CLSI M07; now reference 63), but for this new antimicrobial substance we used a well-cited experimental MIC and NIC calculation protocol since no national or international standard applies.

Reviewer #2

Overall, the identification of Enterococcus species resistant to venom is interesting but the presentation of the data in this manuscript makes the purpose of this study difficult to understand. It almost feels that there are two different stories - one about the microbiome and one about the characterisation of the Enterococcus strain. At the moment there are no clear aims and the manuscript is confusing to read with lots of data in supplements and no clear path to understand what the manuscript is about. The presentation is very long and the results are described in a largely

unnecessary way and should be shortened - in this regard, it almost felt that the results were combined with a discussion, but then there was another discussion on top. The discussion itself does not nicely wrap up the story and contributes to further distraction as e.g., clinical implications that are not necessarily relevant to this study were discussed.

We understand and agree with the reviewer's position that this manuscript feels like two stories- one about the microbiome and one about the characterisation of the *Enterococcus faecalis* strains (2). Indeed our work was carried out in two stages- one describing the microbiota, and one understanding the nature (*E. faecalis* instead of *Staphylococcus*) and adaptation of the most abundant viable aerobes in at least one of the host species.

We feel that reporting on the microbiota in venoms in itself is interesting, because it sets the stage with regards to the evolutionary gradient between envenomation apparatus and venom itself, across which adaptation takes place. However, the natural question that arises is "how do these bacteria manage to survive in venom?" Therefore, it is of equal interest to report upon the mechanisms of adaptation to these extreme environments. We would therefore like to thank the reviewer for their position and, at the editors' discretion, we would therefore consider splitting the manuscript into two papers covering each aspect independently.

To make the aims explicit, we have added a new paragraph at the end of the introduction (page 5, lines 15-21).

In line with the editors and reviewer no.2's recommendation, we have shortened the presentation of results to the extent possible without removing any of the existing supplementary datasets, and rephrased the main text to improve upon clarity. In line with reviewer no. 3's request for additional *E. faecalis* genome analysis, however, we have expanded upon virulence findings across the 9 isolates in the results section (page 14 line 1 – 14).

The discussion section therefore now brings together the evidence in the field that bacteria adapt to venoms, to then colonise envenomation wounds opportunistically during the envenomation injury, which is mechanistically substantiated in this paper at least for *E. faecalis* venom isolates.

In consolidating the clinical implications to the discussion section as requested by reviewer no. 2 (page 16, line 11 - 2), and in line with the interests expressed by reviewer no. 3, we now substantiate more readily how biochemical misdiagnosis could affect clinical management, with recommendations on how to overcome these problems. Given the size of morbidity and mortality burden of snakebite it is our view that highlighting these issues is necessary to improve upon management practice.

Another major limitation about this study and the overall idea of microbes in venom is that the microbial adaptation to venom was only demonstrated in two *E. faecalis* isolates from one snake species - so it is not clear how common microbial adaptation to venom would be in multiple venomous species. The focus is almost exclusively on *N. nigricollis*, so this should be reflected in the manuscript - at the moment, this study, as presented, shows preliminary evidence of their idea, but requires follow up experiments to show whether adaptation is a common phenomenon.

We agree with the reviewer and therefore have shortened the relevant discussion section and explicitly state that adaptation is documented in *E. faecalis* isolates, but it remains unclear to which extent this paradigm extends beyond *N. nigricollis* or *E. faecalis* (page 14, lines 19-23). We then call for further work in this field (page 15, lines 4-6)

Two scorpion species are mentioned but not followed up - it is unclear why they were not discussed?

We thank the reviewer for highlighting the limited analysis of this aspect of our work. Accordingly, we have removed the scorpion data from the manuscript.

There are numerous grammatical errors - too many to mention; the references need to be formatted and updated.

We thank the reviewer for highlighting the need for further proofreading and have incorporated all the recommendations from Reviewer no. 3 and had native English speakers from within the author list review grammatical content.

We have now also revised the manuscript references to the Journal style.

Reviewer #3

1- The title of the paper "Microbial adaptation to venom in snakes and spiders" suggests that different microbes are considered while the paper only describes bacteria. Moreover, not only spiders but snakes and scorpions are studied. Thus the title could be changed to "Bacterial adaptation to venom in snakes and arachnida".

We thank the reviewer for their recommendation and have accordingly modified the manuscript title to "Bacterial adaptation to venom in snakes and arachnida".

2- The term "microbiome" is used extensively throughout the paper without differentiating "microbiome" from "microbiota". These two terms have been clearly defined considering on one side strictly the microbial communities that occupy a given habitat that are defined as the "microbiota", or on the other side the whole system that is defined as "microbiome", including both the microbial communities and their structural elements (nucleic acids, proteins, polysaccharides,..) and metabolites involved in the occupation of a niche (see Berg G. et al 2020, *Microbiome*, 8: 103). The authors have to check the manuscript for the undifferentiated use of "microbiome" and change it to "microbiota" when required. Moreover, the use of the term 'flora' (page 4 line 17, p. 6 line 16, p. 16, line 18) to name the microbial communities in various compartments, is currently out of date and the term "microbiota" should be preferred. Changes have to be made appropriately. For instance page 4 line 17 the title "The snake venom microbiome varies on account of host species and not on account of the oral flora" should be "The snake venom microbiota varies on account of host species and not on account of the oral microbiota", or page 5, lines 16, 17, change "flora" and "microbiome" to "microbiota".

We thank the reviewer for their recommendation and have reviewed and corrected individually all instances where the terms were used incorrectly.

3- The authors have to specify in the paper that they identified all bacterial species, cultivable and non-cultivable, in venoms or oral cavities but that only cultivable bacteria have been considered ("aerobic viability") to assess adaptation to the venom environment.

We thank the reviewer for their recommendation. Accordingly, in the opening sentence of the section concerning viability (page 8, line 5) we explicitly state "*After identifying all cultivable and non-cultivable bacterial species... we next proceeded to examine if cultivable aerobes could be recovered... as an indication of adaptation to venom*". Accordingly, in discussing findings with *B. arietans* we extend statements on the impact of venom handling on microbial viability to "*at least for aerobic bacteria*".

4- The paper is well written, but some sections, especially section "Pangenomic evidence of *E. faecalis* isolate adaptation to venom" and the discussion part could be more concise or more focused.

We thank the reviewer for their recommendation. In line with reviewer no. 2's request and the editor's request, we have reworded this section to reduce its length and for clarity purposes.

We have also moved the analysis on the clinical implications of the *E. faecalis* genomes to the discussion section (page 16, lines 11-2). We have made efforts to reduce the verbosity of the discussion section, focusing now on bringing together the evidence beyond our work pointing to more widespread adaptation of microbes to venoms, and contextualising our findings to the clinical implications of envenomation-derived infections. In line with reviewer no. 3's interests in the virulence of these *E. faecalis* isolates, we feel this section is important given epidemiological evidence and the morbidity and mortality burden of snakebite across resource-limited territories.

5- Materials and methods:

- it is unclear (page 18) how snake venom samples were conserved before lyophilization (-80°C as for spiders?): please specify. This is especially questionable for field-collected samples. Moreover it is suggested line 22 page 6 that venom samples collected from wild animals could have been only air-dried and not conserved at very low temperature and further lyophilized, which very presumably should impact both the venom chemical content and therefore the bacterial strains maintained in this medium. This has to be absolutely clarified and justified.

We thank the reviewer for their highlighting this omission in such a crucial part of the methodological reporting in our manuscript. We have accordingly:

- Expanded Page 18 lines 15-17 to state how snake and spider venoms were preserved between collection and analysis.
- Expanded upon the microbiota results presentation section (page 6, line 22 – page 7 line 2) to explicitly state the unknown provenance of the Latoxan sample, and how

- the air-drying of the wild-caught samples could have been reasonably expected to have a big impact on the 16S profile, but it does not appear to have done so.
- Expanded upon the justification of the handling protocol for venom samples collected in the wild (page 18 line 21 page 19 line 5).

Whilst we agree that the air drying and hyperconcentration of the *B. arietans* wild venoms could have reasonably destroyed any viable bacteria, we note that the Venomtech animal venom did not yield any viable aerobic isolates either. Therefore, in line with the justification stated in the manuscript and the experimental observations presented we feel that the statement restricted to aerobic viability (page 8, line 14) is appropriate given consistency of results.

- it is unclear page 17 lines 18-19 and when comparing to the legend to figures, which samples analyzed are originating from Venomtech and Latoxan and which have been collected from captive or wild animals. Furthermore, it is unclear when commercial lyophilized samples (with unknown caution taken to avoid contamination by external bacteria...) and collected samples were used. This has to be specified very precisely.

We have clarified readability by annotating the relevant entries in Table 1 with L and B1-8, and adding the relevant labels to legend of Figure S3. We have additionally explicitly stated in the methods section (page 18, lines 2-3) which samples were freeze-dried, and which samples were air-dried.

- there is no information about treatments of scorpions and their venom in Materials and methods, why they are considered in Table 1 and in the text; please complete.

We thank the reviewer for highlighting the limited analysis of this aspect of our work. Accordingly, we have removed the scorpion data from the manuscript.

6- While two scorpions have been included in the study, neither conclusions nor discussion are provided about these venoms and their potential bacterial content. Either data related to scorpion venoms have to be described and discussed in the manuscript, or if they do not afford interesting information, they have to be suppressed from the study.

We thank the reviewer for highlighting the limited analysis of this aspect of our work. Accordingly, we have removed the scorpion data from the manuscript.

7- Even if the genes identified in *E. faecalis* and discussed in the paper are trivial and well-known, the functions of the encoded proteins should be specified along the paper to make it more easily understandable by readers that are not strictly involved in the topic, and thus allow broadening the interested scientific community. Otherwise, a list of abbreviations could be provided. In addition or alternatively in certain cases, UniProt codes for the most critical genes identified could be provided.

We thank the reviewer for highlighting the need to expand on the names of the genes discussed (*repA-2*, *gyrA*, *parC*, *macB5*). These, alongside new entries pertaining to the

virulence and bacteriocin genes have now been identified in terms of function (page 10, line 12, 21, 21; page 13, lines 16, 20-23; page 14 lines 1-14)

8- The mammalian origin of strain *E. faecalis* V583 (usually found in soils, waters, food etc and very probably arising from mammalian gastrointestinal tract via feces), which is the closest relative to *E. faecalis* strains from *N. nigricollis* venom has to be specified in the paragraph page 7 lines 11-22.

We thank the reviewer for this recommendation. A statement thereto has been added to the recommended section (page 9 lines 4-5).

9- Resistance of the venom-derived *E. faecalis* strains to the venom antimicrobial content is evaluated in the paper, but pathogenicity of the strains is not examined, while one of the objectives of the paper is to demonstrate that infections associated with envenomations can be due to pathogens contained in the venom. *Enterococcus* spp. are opportunistic pathogens and nothing indicates that the strains isolated from venom samples could be commensals or pathogens. Numerous virulence factors have been identified in *Enterococcus* species and especially in *E. faecalis*. Did the authors investigated this aspect by searching for specific genes encoding virulence factors, such as gelatinase (gelE, fsrB), cytolysin (cyl), aggregation substance, etc (see references Nami Y. et al 2015, Front. Microbiol. 6:782; Hashem Y.A. et al. 2021, Infect Drug Res, 2021:14)? This important information should be added.

We thank the reviewer for this recommendation. Accordingly, we have used VirulenceFinder and manual searching (methods detail added in page 23 lines 23 – page 24 line 4) and findings are presented in the relevant results section, succinctly in line with other requests from both reviewers and without the addition of yet another supplementary table, in page 13 line 23 - page 14 line 14), citing Hashem *et al.* and Nami *et al.*

10- Furthermore, the authors consider the antimicrobial molecules usually described as present in venoms, including antimicrobial peptides, but do not take into account that the antimicrobial peptides produced by the bacteria (bacteriocins) compete against other closely related strains in the same niche and thus could also contribute to the selection of the *E. faecalis* isolates present in the venom by killing other bacteria genera close to the producer thanks to their bacteriocins. The authors mention briefly the production of bacteriocin 41 (Bac41), but many other bacteriocins have been identified from *Enterococcus* strains (mainly enterocins L50A/L50B, A, B, P, Q, X, AS-48; bacteriocins 32, 43, 51). Did the authors searched for the genes encoding such well-known bacteriocins? The information has to be added and the role of the bacteriocins in adaptation of bacteria to venom conditions has to be discussed.

We thank the reviewer for bringing to our attention this aspect of opportunistic dominance for the *E. faecalis* strains identified in *N. nigricollis* venoms in our study. To this end we have used the well cited bacteriocin miner BAGEL4 to identify relevant genes in our isolates and included this information in the results section (page 10, line 19 – page 11 line 3).

11- Figures:

- Figure 1: Why is B3 absent from Figure 1B, while the legend indicates "The 8 wild (red dots B1-B8)"?

We thank the reviewer for spotting this reporting oversight. An explanatory note has been added to Figures 1 and S3 on the ~100x lower read depth obtained from sample B3, leading to its exclusion from these analyses.

- Figure 3 is unclear: B and D are lacking.

Labels B and D have been added to the figure.

12- Minor points

- Page 5, title last line change "A fifth of the *N. nigrocollis* venom microbiome is distinct to that of fangs" to "A fifth of the *N. nigrocollis* venom microbiota is distinct from that of fangs"

- Typing mistakes - page 7 line 4, in "*Staphylococcus spp.*", "*spp.*" should not be in italics; page 11: line 9 change "Biotin transporter" to "biotin transporter", "Bacterial regulatory" to "bacterial regulatory", "gntR" to "GntR"; line 15 change "L-Amino acid" to "L-amino acid"; page 12: line 4 change "was" to "were", line 7 change "L-Amino acid" to "L-amino acid", line 9 change "l-amino acid" to "L-amino acid"; page 13 "*macB5*" should be in italics; page 37, Table 1 footnote: use the greek letter instead of "ul" for microliter; page 38 Table 2: change "iqil" to "iqiL"; Legend to figure 3: line 10, *N. nigricollis* should be in italics.

All changes made as requested.

Bacterial adaptation to venom in snakes and arachnida.

Response to Reviewers: Point by point changes (second revision)

Reviewer #3

1- The Abstract needs being improved by using a past tense (preterit) rather than the present for describing what has been done or observed: Change "observe" to "observed" line 19 page 2, "evidence" to "evidenced" line 20 page 2, "identify" to "identified" line 2 page 3, and "an additional 45 genes" to "45 additional genes" line 3 page 3.

Done

2- Introduction page 5 line 18, change "examine" to "examined" and "employ" to "employed"

Done

3- Page 11, line 11 change "group groups" to "groups; line 13, change "an additional 4 genes" to "an additional 4 gene set"

Done

4- Page 12, line 6 change "faecailis" to "faecalis"

Done

5- Page 13, line 15 change "antimicrobial" to "antibiotic" to better express that conventional antibiotics are considered here.

Done

6- Page 15 line 12 change "was additionally" to "were additionally"

Done

7- List of references:

-Ref 46 and 47: author names are lacking;

Fixed – our apologies; this appears to be a persistent Mendeley error despite correct database entries.

-Ref 23: two years are indicated 2003 and (1979): suppress 1979

Fixed – database input error

-Ref 44 is incomplete (what does Toxicon X4 means?)

Fixed – database input error again.

-Names of bacteria, snakes etc should be in italics; conform to the journal requirements but if capital letters are used for each name in the titles this should be homogeneous over the list; similar, the names of the journals should be abbreviated using the conventional abbreviations and not alternatively full names/abbreviations.

We have revised line by line all references to ensure maximal compliance to journal standards and consistency in formatting, resulting in changes in Title Case to Sentence case, italicisation of all genus and species names, and database entry corrections for a number of entries.

April 14, 2022

Dr. Sterghios A Moschos
Northumbria University
Applied Sciences
Ellison building
Ellison Place
Newcastle Upon Tyne, Tyne and Wear NE1 8ST
United Kingdom

Re: Spectrum02408-21R2 (Bacterial adaptation to venom in snakes and arachnida)

Dear Dr. Sterghios A Moschos:

Thank you for your fast resubmission.

Your manuscript has been accepted, and I am forwarding it to the ASM Journals Department for publication. You will be notified when your proofs are ready to be viewed.

Sincerely,

Olaya Rendueles Garcia
Editor, Microbiology Spectrum

Journals Department
Supplemental tables S1-S10: Accept
Supplemental Figures S1-S9 and Tables S1-4, S8: Accept